# WHAT DO LARGE NETWORKS MEMORIZE?

## ABSTRACT

The success of modern neural models has prompted renewed study of the connection between *memorisation* and *generalisation*: such models typically generalise well, despite being able to perfectly fit ("memorise") completely random labels. To more carefully study this issue, Feldman (2019); Feldman & Zhang (2020) provided a simple stability-based metric to quantify the degree of memorisation of a specific training example, and empirically computed the corresponding memorisation profile of a ResNet model on image classification benchmarks. While an exciting first glimpse into how real-world models memorise, these studies leave open several questions about memorisation of practical networks. In particular, how is memorisation affected by *increasing* model size, and by *distilling* a large model into a smaller one? We present an empirical analysis of these questions on image classification benchmarks. We find that training examples exhibit a diverse set of memorisation trajectories across model sizes, with some samples having *increased* memorisation under larger models. Further, we find that distillation tends to *inhibit memorisation* of the student model, while also improving generalisation. Finally, we show that other memorisation measures do not capture such properties, despite highly correlating to the stability-based metric of Feldman (2019).

## 1 INTRODUCTION

Statistical learning is conventionally thought to involve a delicate balance between *memorisation* of training samples, and *generalisation* to test samples (Hastie et al., 2001). However, the success of modern *overparameterised* neural models challenges this view: such models have proven successful at generalisation, despite having the capacity to memorize, e.g., by perfectly fitting completely random labels (Zhang et al., 2017). Indeed, in practice, such models typically *interpolate* the training set, i.e., achieve zero misclassification error. This has prompted a series of analyses aiming to understand why such models can generalise (Bartlett et al., 2017; Brutzkus et al., 2018; Belkin et al., 2018; Neyshabur et al., 2019; Bartlett et al., 2020; Wang et al., 2021).

Recently, Feldman (2019) established that in some settings, memorisation may be *necessary* for generalisation. Here, "memorisation" of a sample is defined via an intuitive stability-based notion, where the high memorisation examples are the ones that the model can correctly classify only if it they are present in the training set (see Equation 1 in §2). A salient feature of this definition is that it allows the level of memorisation[1] of a training sample to be *estimated* for practical neural models trained on real-world datasets. To that end, Feldman & Zhang (2020) studied the memorisation profile of a ResNet model on standard image classification benchmarks. While an exciting first glimpse into how real-world models memorise, this study leaves open several questions about the nature of memorisation as arising in practice. We are particularly interested in two questions:

— *Model size and memorisation.* Increasing model size (e.g., depth of a ResNet) has a well-documented effect of (unsurprisingly) improving training accuracy, and (surprisingly) test accuracy as well (Neyshabur et al., 2019). It is unclear what impact model size has on memorisation, however; while larger models have more memorisation *capacity*, do they make judicious use of this and memorise *fewer*, more informative samples than smaller models?

— *Distillation and memorisation.* While the study of memorisation in large neural models is fascinating, its practical relevance is stymied by a basic fact: such models are typically inadmissible

---

[1]From hereon in, unless otherwise noted, we shall use "memorisation" to refer to the stability-based notion of Feldman (2019); see Equation 1.

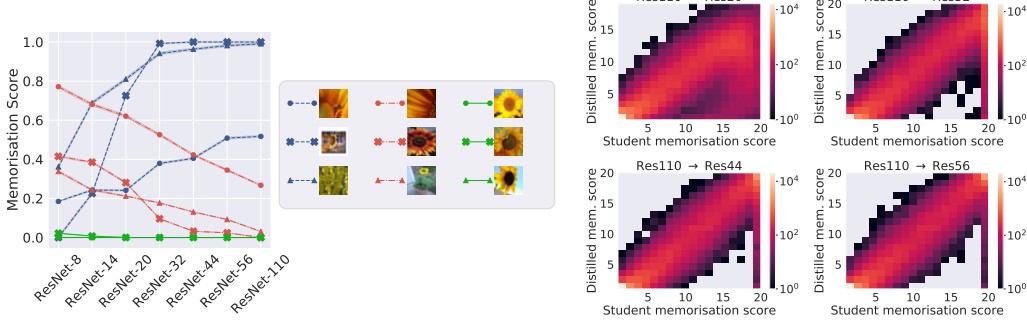

(a) Example of memorisation trajectories over depth. (b) Change in memorisation under distillation.

Figure 1: Illustration of how memorisation (in the sense of Equation 1) evolves with ResNet model depth on CIFAR-100 in one-hot training (left) and distillation (right) setups. In the left plot, we show that *training examples exhibit a diverse set of memorisation trajectories* across model depths: fixing attention on training examples belonging to the sunflower class, while many examples unsurprisingly have fixed or decreasing memorisation scores (green and red curves), there are also examples with increasing memorisation (blue curves). Typically, easy and unambiguously labelled examples follow a fixed trend, noisy examples follow an increasing trend, while hard and ambiguously labelled examples follow either an increasing or decreasing trend; in §3 we discuss their characteristics in more detail. In the right plot, we show that *distillation inhibits memorisation of the student*: each plot shows the joint density of memorisation scores under a standard model, and one distilled from a ResNet-110 teacher. As the gap between teacher and student models becomes wider, samples memorised by the student see a sharp decrease in memorisation score under distillation (see the vertical bar at right end of plot). Notably, distillation does not affect other samples as strongly.

for real-world settings with constraints on latency and memory. Instead, such models are typically compressed via *distillation* (Bucilă et al., 2006; Hinton et al., 2015), a procedure that involves fitting a smaller "student" model to the predictions of a "teacher". This raises a natural question: *how much of the teacher model's memorisation is transferred to the student*?

In this paper, we take a step towards comprehensively exploring the memorisation behavior of modern neural networks, and in the process contribute to progress on answering the aforementioned questions. Fixing our attention on the elegant stability-based notion of Feldman (2019), we study how memorisation varies as capacity of common model architectures (ResNet, MobileNet) increases, with and without distillation. We have four main findings: first, the distribution of memorisation scores becomes increasingly *bi-modal* with increased model size, with relatively more samples assigned either a very low *or* very high score. Second, *training examples exhibit a diverse set of memorisation trajectories* across model sizes, and in particular, some examples are *increasingly* memorised by larger models. Third, while distillation largely preserves the student model's memorisation scores, it tends to *inhibit memorisation* in the student. Finally, we analyze computationally tractable measures of memorisation which have been shown in previous work to highly correlate to the stability-based notion of memorisation score, and show that surprisingly they lead to significantly different properties.

To summarise, our contributions are:

(i) we present a quantitative analysis of how the degree of memorisation of standard image classifiers varies with model complexity (e.g., depth or width of a ResNet). Our main findings are that increasing the model complexity tends to make memorisation more *bi-modal*; and that there are examples of varying memorisation score trajectories across model sizes, including those where memorisation *increases* with model complexity.

(ii) we then present a quantitative analysis of how distillation influences memorisation, and show that it tends to inhibit memorisation, particularly of samples that the one-hot (i.e., non-distillation) student memorises.

(iii) we conclude with experiments using computationally tractable measures of memorisation and example difficulty, based on behaviour across training steps (Jiang et al., 2021a), and behaviour across model layers (Baldock et al., 2021). We find that although these measures strongly correlate with stability-based memorisation, they do not capture key trends seen in the latter.

We remark that the aim of this work is *not* in proposing new methods, or memorisation measures, but rather to more systematically analyse the behaviour of existing memorisation measures.

## 2 BACKGROUND AND RELATED WORK

The term "memorisation" is often invoked when discussing supervised learning problems, but with several distinct meanings (cf. Table 1 in Appendix). A classical usage of the term refers to models that simply construct a fixed *lookup table*, *viz.* a table mapping certain keys to targets (e.g., labels); predictions are made by simply identifying a given row of this table, and outputting the associated target. This includes the classic $k$-nearest neighbour ($k$-NN) algorithm (Devroye et al., 1996), and a recent construct of Chatterjee (2018). Noting that the 1-nearest neighbour algorithm is capable of *interpolating* the training data (i.e., perfectly predicting every training sample), some works instead use "memorisation" as a synonym for interpolation (Zhang et al., 2017; Arpit et al., 2017; Stephenson et al., 2021). A more general definition is that "memorisation" occurs when the training error is lower than the best achievable error (or *Bayes-error*) (Bubeck et al., 2020; Cheng et al., 2022).

While each of these notions imply "memorising" the *entire* training set, one may also ask whether a *specific* sample is memorised. One intuitive formalisation of this notion is that the model predictions change significantly when a sample is removed from training (Feldman, 2019; Jiang et al., 2021b), akin to the notion of algorithmic stability (Bousquet & Elisseeff, 2002). More precisely, consider a training sample $S$ comprising $N$ samples, drawn *i.i.d.* from some distribution $D$ over labelled inputs $(x, y) \in \mathcal{X} \times \mathcal{Y}$. A learning *algorithm* is some randomised function $\mathsf{A}(\cdot; S) \colon \mathcal{X} \to \mathcal{Y}$, where the randomness is, e.g., owing to random initialisation, ordering of mini-batches, and stochasticity in parameter updates. The *memorisation score* of a sample $(x, y) \in S$ is then (Feldman, 2019):

$$\mathsf{mem}(x, y; S) = \mathbb{P}(y = \mathsf{A}(x; S)) - \mathbb{P}(y = \mathsf{A}(x; S - \{(x, y)\})). \tag{1}$$

Here, $\mathbb{P}(\cdot)$ considers the randomness in the learning algorithm. Intuitively, this is the excess classification accuracy on the sample $(x, y)$ when it is *included* versus *excluded* in the training sample. Large neural models can typically drive the first term to 1 for *any* $(x, y) \in S$ (i.e., they can interpolate any training sample); however, some $(x, y)$ may be very hard to predict when they are not in the training sample. Such examples may be considered to be "memorised", as the model could not "generalise" to these examples based on the rest of the training data alone.

Despite its strengths, Equation 1 has an obvious drawback: it is prohibitive to compute in most practical settings. This is owing to it naïvely requiring that we retrain our learner at least $N$ times, with *every* training sample excluded once; accounting for randomness in $\mathsf{A}$ would require further repetitions. Feldman & Zhang (2020) provided a more tractable estimator, wherein for fixed integer $M$, one draws $K$ *sub-samples* $\{S^{(k)}\}_{k \in [K]}$ uniformly from $\mathcal{P}_M(S)$, the set of all $M$-sized subsets of $S$. For fixed $(x, y)$, let $K_{\mathrm{in}} \doteq \{k \in [K] \colon (x, y) \in S^{(k)}\}$, and $K_{\mathrm{out}} \doteq \{k \in [K] \colon (x, y) \notin S^{(k)}\}$ denote the sub-samples including and excluding $(x, y)$ respectively. We then compute

$$\widehat{\mathsf{mem}}_{M,K}(x, y; S) = \frac{1}{|K_{\mathrm{in}}|} \sum_{k \in K_{\mathrm{in}}} [\![y = \mathsf{A}(x; S^{(k)})]\!] - \frac{1}{|K_{\mathrm{out}}|} \sum_{k \in K_{\mathrm{out}}} [\![y = \mathsf{A}(x; S^{(k)})]\!]. \tag{2}$$

This quantity estimates $\mathsf{mem}(x, y; S)$ to precision $\mathcal{O}(1/\sqrt{K})$. Jiang et al. (2021a) considered a closely related *consistency score* (or *C-score*), defined as

$$\mathsf{cscore}(x, y; S) = \mathbb{E}_M \left[ \mathbb{E}_{S' \sim \mathcal{P}_M(S)} \left[ \mathbb{P}(y = \mathsf{A}(x; S' - \{(x, y)\})) \right] \right].$$

When $M$ is drawn from a point-mass, this is the second term in $\widehat{\mathsf{mem}}_{M,K}(x, y; S)$. Note that the first term in $\widehat{\mathsf{mem}}_{M,K}(x, y; S)$ is typically 1 for overparameterised models, since they are capable of interpolation. Jiang et al. (2021a) proposed effective *proxies* for the C-score, which study the model behaviour across training steps. Specifically, suppose we have a model that is iteratively trained for steps $t \in \{1, 2, \dots, T\}$, where at $t$-th step, the model produces a probability distribution $\hat{p}^{(t)} \colon \mathcal{X} \to \Delta(\mathcal{Y})$ over the labels. Jiang et al. (2021a) established that the metric

$$\mathsf{cprox}(x, y; S) = \mathbb{E}_t \left[ \hat{p}_y^{(t)}(x) \right] \tag{3}$$

can correlate strongly with (the point mass version of) $\mathsf{cscore}(x, y; S)$. We defer to Appendix A a discussion of other related notions of *example difficulty*.

**Prior empirical analyses of memorisation**. Several works have studied the *interpolation* behaviour of neural models as one varies model complexity (Zhang et al., 2017; Neyshabur et al., 2019). Empirical studies of memorisation in the sense of fitting to random (noisy) labels was conducted by Arpit et al. (2017); Gu & Tresp (2019). These works demonstrated that real-world networks tend to fit "easy" samples first, and exhibit qualitatively different learning trajectories when presented with clean versus noisy samples; this was used to argue that networks tend not to simply memorise real-world datasets. Zhang et al. (2020) provided an elegant study of the interplay between memorisation and generalisation for a regression problem, involving learning either a constant or identity function.

Feldman & Zhang (2020) studied memorisation in the sense of the stability-based memorisation score in Feldman (2019) (cf. Equation 1), by quantifying the influence of each training example on different test examples. Based on these, they identified a subset of test examples for which the model significantly relies on the memorized (in the stability sense) training examples to make correct predictions. While the direct inspiration for our study, these experiments were for a single architecture on CIFAR-100 and ImageNet, and did not consider the impact of model distillation. Zheng & Jiang (2022) confirmed the theory of Feldman (2019) holds in the context of NLP as well, by studying memorisation of BERT models using a related notion of *influence functions* (Koh & Liang, 2017). Elangovan et al. (2021) quantified how much *data leakage* affects a BERT model, while Tänzer et al. (2021) showed that BERT models memorise random labels only in later stages of training.

## 3  How does memorisation evolve with model size?

We now seek to quantify (via Equation 1) how memorisation varies with neural model capacity.

**Setup and scope.** Quantifying the nature of "memorisation" requires picking a suitable definition of the term. Owing to its conceptual simplicity and intuitive alignment with the term "memorisation", we employ the stability based memorisation score of Feldman (2019), per Equation 1. For computational tractability, we employ the approximation to this score from Feldman & Zhang (2020), per Equation 2. This reduces the computational burden of estimating $\mathrm{mem}(x, y; S)$, but does not eliminate it: for *each* setting of interest, we need to draw a number of independent data sub-samples, and train a fresh model on each. This necessitates a tradeoff between the breadth of results across settings, and the precision of the memorisation scores estimated for any individual result. We favour the former, and estimate $\widehat{\mathrm{mem}}_{M,K}(x, y; S)$ via $K = 400$ draws of sub-samples $S' \sim \mathcal{P}_M(S)$, with $M = \lceil 0.7N \rceil$.

With the above setup, we examine a simple question: *how is memorisation influenced by model capacity?* We aim to answer this empirically. Specifically, for a range of standard image classification datasets — CIFAR-10, CIFAR-100, and Tiny-ImageNet — we empirically quantify the memorisation score as we vary the capacity of standard neural models, based on the ResNet (He et al., 2016b;c) and MobileNet-v3 (Howard et al., 2019b) family (cf. Appendix B for precise settings).

**How do we expect model size to affect memorisation?**. Before any empirical analysis, it is worth pausing to consider how one might intuitively expect model size to affect memorisation. Recall that in image classification with neural models, with increasing model capacity we see improving *generalisation* on the test set (cf. Table 3 in Appendix). Consequently, a natural hypothesis is that memorisation of individual training samples might initially increase with model capacity (as a weak model may have insufficient capacity to memorise), but should eventually saturate or reduce (as more expressive models generalise better). We also conjecture how we expect memorisation of an *individual* example to evolve as model size increases. For most examples, we expect that memorisation should decrease or be constant, since the larger models generalise better and thus can model even complex samples. We now assess whether this intuition is borne out empirically.

**Larger models → memorisation scores reduce on average.** In Figure 2(a), we visualise how ResNet model depth influences the memorisation score (Equation 1) on CIFAR-100. Specifically, for each ResNet model, we report the *average* memorisation score across all training samples as a coarse summary. Consistent with our hypothesis above, this score increases up to depth 20, and then starts *decreasing* (albeit slightly). This phenomenon may be easily understood by breaking down the two terms used to compute the memorisation score in Equation 1: the *in-sample accuracy*, and the *out-of-sample accuracy*. As expected, both quantities steadily increase with model depth. The increasing memorisation score up to depth 20 can be explained by the former increasing *faster* than

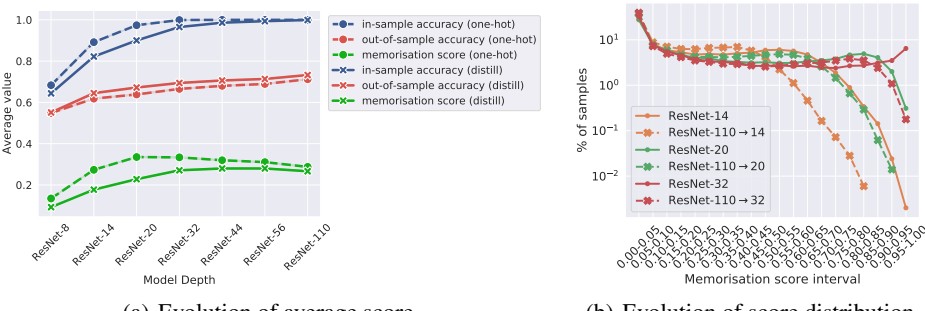

(a) Evolution of average score.      (b) Evolution of score distribution.

Figure 2: Memorisation scores from distilled and one-hot models on CIFAR-100 training samples. We see from the left subfigure that the average memorisation score steadily *increases* with depth up till the ResNet-20, and steadily *decreases* afterwards. The increase up to a certain depth can be explained by the in-sample accuracy increasing faster than out-of-sample accuracy, until the point where the former is ∼ 100%. The average memorisation score is reduced by distillation across model depths, which can be attributed to both the reduced in-sample accuracy and increased out-of-sample accuracy. The right subfigure depicts this phenomenon in finer granularity: as the model depth increases, relatively more samples are highly memorised (notice the hump in the graphs is being progressively pushed to the right); At the same time, notice how the average memorisation goes down with higher depths. We also observe that distillation reduces memorisation scores across model depth. Overall we find a growing *bi-modality* of the distribution of scores with increasing depth.

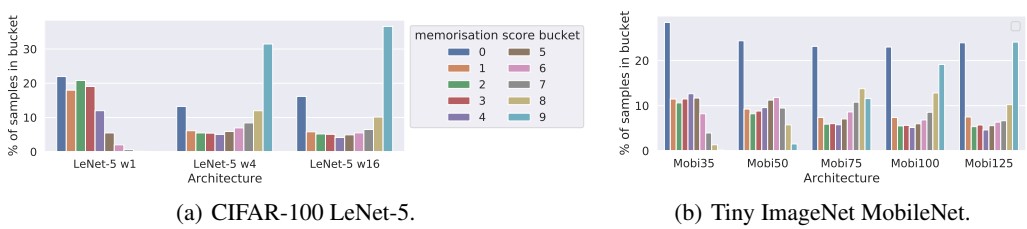

(a) CIFAR-100 LeNet-5.      (b) Tiny ImageNet MobileNet.

Figure 3: Distributions of memorisation scores across datasets and model architectures. We divide the range of memorisation scores into 10 equally-spaced memorisation score buckets. We consider LeNet-5 (LeCun et al., 1998) models on CIFAR-100 training set and MobileNet models on Tiny Imagenet of varying sizes. We achieve multiple architecture versions for LeNet-5 by varying the widths: w*k* denotes a model where width is scaled *k* times compared to the original LeNet-5. We find an increasingly *bi-modal* distribution as the width increases.

the latter up to this point; beyond this point, the in-sample accuracy saturates at 100%, while the out-of-sample accuracy keeps increasing. Thus, necessarily, the memorisation score starts to drop.

**Larger models → memorisation scores become more bi-modal.** The preceding result only considered the *average* memorisation score for a given model. But how does the *distribution* of memorisation scores vary with model capacity? To this end, Figure 2(b) plots precisely this distribution for each model in consideration. We observe that the memorisation scores tend to be *bi-modal*, with most samples' score being close to 0 or 1. This is in agreement with observations made in Feldman & Zhang (2020) for a *fixed* ResNet architecture, on both CIFAR-100 and ImageNet.

More interestingly, this bi-modality is *exaggerated with model depth*: larger models have a higher fraction of samples with both memorisation score close to 0 and 1. The former is consistent with improved generalisation of larger models, and is thus unsurprising. On the other hand, the latter denotes an increasing generalisation gap on a subset of training points, which does *not* harm the test set generalisation performance. Instead, as model capacity increases, we typically see monotonically increasing test set performance (cf. Table 3 in the Appendix). In Figure 3 we present results for

LeNet-5 on CIFAR-100 and MobileNet on Tiny Imagenet, and demonstrate that our observations hold across architecture families and datasets.

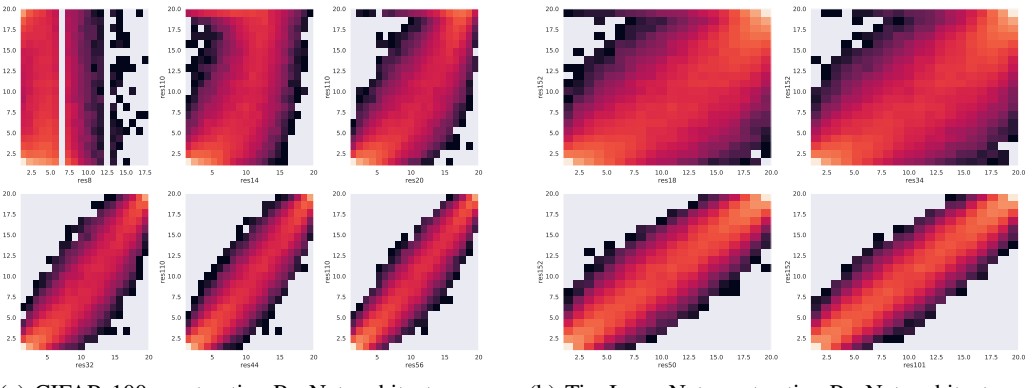

(a) CIFAR-100: contrasting ResNet architectures.  (b) TinyImageNet: contrasting ResNet architectures.

Figure 4: Contrasting per-example memorisation scores across architectures in different setups. As the difference in depths becomes larger, the highly memorised examples according to a small model become *less* memorised by the large model (note the high density in the lower-right part of the plot). On the other hand, smaller models tend to have examples which get assigned a high memorisation score by a large model (note the horizontal bar at the top of a figure).

**Larger models → some samples are memorised *more*.** Given the previous observation of the increasing bi-modality with model depth, we now study how exactly the memorisation score of each example shifts across model depths. In Figure 4, we show how memorisation scores evolve with depth on a *per-example* basis. An unsurprising observation is that memorisation scores in most cases do not significantly change across depths. Beyond this, we can see that as the difference in depths becomes larger, the highly memorised examples according to a small model become *less* memorised by the large model (note the high density in the lower-right part of the plot). Most interestingly, in all cases, there is a non-trivial fraction of samples whose memorisation score *increases* with model size; note the horizontal bar at the top of Figure 4, and the bump in the right side of the marginals depicted in Figure 2(b). Increasing memorisation implies a *decreasing* out-of-sample accuracy on such points (as the in-sample accuracy monotonically increases with model depth). This is perhaps surprising, since one expects increasing model depth to improve generalisation (Neyshabur et al., 2019).

To sum up, we observed that there are examples with both *increasing* and *decreasing* memorisation with model depth, as denoted by the out-of-diagonal high density regions in Figure 4. Next, we inspect the specific examples exhibiting such varying trajectories.

**Examining samples memorised more/less by larger models.** Next, we investigate examples exhibiting varying memorisation trajectories over model size. We are particularly interested in studying examples which increase and decrease in memorisation. Recall that we conjectured different trajectories of memorisation depending on the example: for the examples which are easy across model depths, we expect a low, stable memorisation; for the mislabeled examples a high, increasing memorisation; and for the hard but unambiguous examples, an increasing trajectory of memorisation.

In Figure 1(a), we report the average memorisation score as a function of depth for examples from the `sunflower` class exhibiting varying trajectory patterns: increasing, decreasing and constant. Also, in Table 10 (Appendix) we report predictions for the shown examples. We first notice that the least-changing memorisation examples are *easy and unambigous*. Further, amongst these examples, the peak memorisation score tends to be low (i.e., such samples tend to consistently not be memorised); indeed, when measuring the change in memorisation by the difference from ResNet-110 and ResNet-20 models, among the least changing 1% examples the average memorisation (as measured on ResNet-110) is 0.06, and among least changing 10% examples is only 0.33.

Next, we note that both the *increasing* and *decreasing* examples are hard and ambiguous. The *decreasing* examples are arguably hard but labeled correctly. On the other hand, the *increasing*

examples are often multi-labeled (e.g., the first *increasing* example containing a bee) or mislabeled (e.g. the third *increasing* example), such that a human rater could be unlikely to label these images with the dataset provided label. This parallels the predictions from ResNet-110 for these examples, which arguably are no less reasonable than the label as presented in the dataset.

This can be intuitively explained by breaking down the *Feldman memorisation* score into in-sample and out-of-sample accuracy components. For the examples which are easy across model depths, we find a low and not changing memorisation as the in-sample accuracy should be low and the out-of-sample accuracy should be low across depths. Next, for the mislabeled examples in the data, we find a high or even increasing memorisation, as with increasing depth the *correct* label, which disagrees with the *noisy* label in the data, is being recovered better by the deeper models, and so the out-of-sample accuracy increases or remains high. Finally, for the hard but unambiguous examples, we find an increasing trajectory of memorisation, as the deeper models gradually get better at generalisating to that point, which corresponds to the out-of-sample accuracy for that example increasing, and in turn, the memorisation score decreasing.

To conclude, we indeed find examples aligned with our hypothesis, i.e., the easy and unambigous images contributing to the *decreasing* examples. However, we also find ambigous and mislabeled images yielding the *increasing* examples. More surprisingly, we also find ambiguous points among the *decreasing* examples. We discuss this in more detail in Appendix D.1, where we inspect more examples and further elaborate on our observations.

**Summary.**  We have found that, contrary to our initial hypothesis, the empirical behaviour of memorisation under model size is nuanced: there are some samples which are memorised less by larger models (as expected), *but* there are also those for which the opposite holds. These reflect the underlying "difficulty" of the example and the sample's label. As we will see in §5, some of the other existing notions of sample difficulty which have been shown to correlate with memorisation score do not capture its properties we discussed. Most notably, we will show how these scores exhibit unimodal score distributions, while the above memorisation score exhibits a bi-modal distribution.

## 4    DOES DISTILLATION TRANSFER MEMORISATION?

Large neural models have shown impressive predictive performance on challenging tasks in vision and NLP. However, in practical settings, it is often infeasible to deploy such models. It is instead more common to *compress* these into more tractable models, with one popular strategy being *distillation* (Bucilă et al., 2006; Hinton et al., 2015). Here, one feeds a large ("teacher") model's predicted probability *distribution* over labels as the prediction targets for a small ("student") model. Compared to training the latter from scratch, distillation can provide significant performance gains; these are informally attributed to distillation performing "knowledge transfer". Knowledge distillation has been succesfuly applied across many applications, including: computer vision (Beyer et al., 2021), language modeling (Sanh et al., 2019), information retrieval (Lin et al., 2021), machine translation (Zhou et al., 2020), and ads recommendation (Anil et al., 2022; Liu et al., 2022).

While distillation has been shown to yield significant performance gains on average, *train accuracy* has been shown to be systematically harmed (Cho & Hariharan, 2019). Previous work also showed how distillation can lead to worsened *accuracy* on a subset of hard examples (Lukasik et al., 2021). In a related study, model compression has been shown to harm *accuracy* on tail classes (Hooker et al., 2019). Given these observations, it is natural to ask what effect such distillation has on memorisation of the resulting distilled models. To assess the impact of distillation on memorisation, we consider knowledge distillation as conducted using logit matching, and where the teacher is trained on the same sub-sample as the student for estimating the memorisation scores per Equation 2. We provide the hyperparameter details in Appendix B.

**Distillation inhibits memorisation.**  We begin by investigating what happens to average memorisation under distillation. Distillation is known to reduce the train accuracy compared to the one-hot models, while increasing the test accuracy (Cho & Hariharan, 2019); in Table 4 (Appendix), we report the train and test accuracies. From this, one could expect distillation to inhibit memorisation. In Figure 2, we illustrate the difference in distributions of memorisation scores across models trained on the ground truth labels (which we call *one-hot training*) and the models distilled from a ResNet-110

teacher model. As expected, we find that distillation tends to *reduce* the number of memorised samples. We also plot in Figure 2 the decomposition of memorisation into in-sample and out-sample accuracies for the one-hot and distilled models. We find that across all model depths, the in-sample accuracy becomes lower and out-of-sample accuracy becomes higher under distillation. This parallels the observation that train accuracy lowers and test accuracy increases under distillation.

In Figure 14 (Appendix), we report how memorisation changes when varying the teacher size. We find that across different student and teacher models, memorisation is inhibited. Interestingly, as the *student* depth varies, the memorisation pattern changes significantly, while with varying the *teacher*, there is no significant change in how memorisation changes between the distilled and one-hot models.

**Distillation inhibits memorisation of highly memorised examples.** We next turn to analysing the distribution of per-example change in memorisation under distillation. In Figure 1(b), we report the joint density of memorisation scores under a standard model, and one distilled from a ResNet-110 teacher. We can see that distillation inhibits memorisation particularly for the examples highly memorised by the one-hot model, and especially when the teacher-student gap is wide. Interestingly, none of the examples with small memorisation score from the one-hot model obtain a significant *increase* in memorisation from distillation.

In Figure 12 (Appendix), we plot trajectories of memorisation over architecture depths for examples depicted in Figure 1. We find that memorisation is overall lowered especially for the challenging and ambigous examples, which often get high memorisation score value by either the small or large models. Notice how this is consistent with the finding from Figure 1(b) of the reduction of memorisation for particularly highly memorised examples. At the same time, none of the low memorisation examples changes their trajectory. This is denoted by the area in the upper left corner of each plotted matrix, which corresponds to count 0.

## 5    DO OTHER MEMORISATION SCORES BEHAVE SIMILARLY?

The preceding analysis has sought to shed light on the memorisation patterns of large neural models, building on the stability-based notion of memorisation in Equation 1. Any attempt to extend this analysis to larger datasets and models faces a fundamental stumbling block: even with the approximation of Equation 2, estimating this score imposes a non-trivial computational burden. It is thus tempting to consider the viability of other, computationally lighter notions of memorisation. To this end, we now consider two such metrics — the C-score proxy and prediction depth — that may also be (informally) tied to memorisation. Interestingly, we find that such metrics do *not* exhibit the same diversity of memorisation behaviour as the stability-based notion. Thus, quantifying memorisation of large models either requires paying a computational price, or devising new metrics.

**The proxy scores.** The C-score proxy cprox (Equation 3) was proposed in Jiang et al. (2021a) as a computationally efficient alternative to the C-score, a metric closely related to the stability-based memorisation score of Equation 1. Indeed, Jiang et al. (2021a) found this measure to have high *correlation* with the C-score, while cautioning that it should not be interpreted as an *approximation* to more fine-grained characteristics of the latter. The C-score also correlates with the *prediction depth* (Baldock et al., 2021), which computes model predictions at intermediate layers, and reports the earliest layer beyond which all predictions are consistent. These predictions were made using a $k$-NN classifier in Baldock et al. (2021); in Appendix F, we explain why using a *linear* classifier is also admissible, and may even be *preferable*. As demonstrated in Baldock et al. (2021), prediction depth offers an appealing balance between being well correlated with other metrics (e.g., it forms a lower bound on the stability-based memorisation score), and being computationally efficient (it only requires training a single model, with re-runs only to estimate standard deviations).

**Correlation to the memorisation score.** In Figure 19 (Appendix), we provide heatmaps of the joint density of these measures against stability-based memorisation. We see that cprox has a visually strong relationship to the latter, and indeed the two have a high degree of correlation: e.g., for the ResNet-56 model, the correlation coefficient is $0.91$. We also find a correlation of prediction depth to memorisation score, albeit to a smaller extent. In particular, the correlation coefficient is $0.72$ for the ResNet-56 model, and there are visually regions of small density outside of the diagonal.

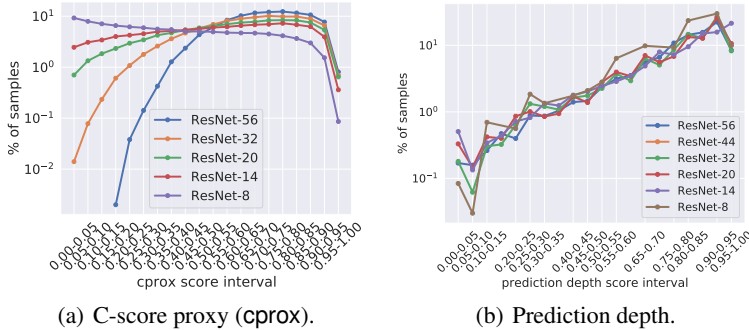

(a) C-score proxy (cprox).    (b) Prediction depth.

Figure 5: Marginal distribution of cprox (left) and prediction depth (right) over the CIFAR-100 training set across ResNet architectures. The marginals are unimodal, unlike the bi-modal distributions for stability-based memorisation scores (see Figure 2(b)).

**Unimodal marginal distribution.** Going beyond correlation, however, we find in Figure 5 that the distributions of cprox and prediction depth scores have markedly different characteristics to stability-based memorisation: the former are *unimodal*, with most samples having a high score value (for both cprox and prediction depth). Recall that by contrast, stability-based memorisation scores exhibit a bi-modal distribution (Figure 2(b)), with this phenomenon exaggerated with increasing model depth. Consequently, we do not observe many of the unexpected trends exhibited by stability-based memorisation, e.g., the possibility of cprox or prediction depth systematically *decreasing* for some samples as depth increases. Indeed, we find that scores do not significantly change across architectures, as can be seen in Figure 18 (Appendix). This is in stark contrast to the stability-based memorisation score, which exhibits large score changes per example when considering architectures of large depth gap. In line with Anscombe (1973), our finding illustrates that while the correlation between two distributions may be high, the nature of the distributions themselves can be significantly distinct. Thus, we conclude it is important to be cautious about conclusions drawn from a high correlation between various memorization scores and their proxies, as they may in reality be capturing different properties of data.

## 6 Discussion and future work

We have seen how increasing model capacity and knowledge distillation both tend to lower memorisation on average. Intriguingly, a distinct line of work has studied the impact of capacity and distillation on *adversarial robustness*: Madry et al. (2018) shows how model robustness increases with capacity, while Papernot et al. (2016) showed how distillation can lead to improved adversarial robustness. It is thus natural to ask how adversarial robustness relates to memorisation. As a first step, we note that intuitively, adversarially robust models ought to produce decision regions which can better generalise to even challenging examples excluded from the train set. We demonstrate this intuition on a toy example in Section G in the Appendix, where a smaller model is less robust and at the same time yields a higher memorisation score for a challenging example.

Sanyal et al. (2022); Paleka & Sanyal (2022) show how interpolating noisy training examples can lead to a higher adversarial error. In our analysis, we found noisy examples to have an *increasing* trajectory of memorisation, thus hinting that models of increasing capacity may be increasingly more robust against fitting to such noisy labels. It will be of interest for future work to systematically analyse noisy training examples through the lens of stability-based memorisation.

As our analysis focussed on image classification settings, a natural question is whether one observes similar trends in NLP tasks. Building on the work of Carlini et al. (2022), which computed stability-based memorisation for a single T5 model, it would be of interest to assess how model size and distillation behave in this context. Another interesting direction would be the study of techniques to improve distillation by weighting samples according to their memorisation score. Finally, contrasting the memorisation behaviour of overparameterised models to sparsely-activated mixture of expert models (Shazeer et al., 2017) would also be of interest.

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

## A  ADDITIONAL RELATED WORK

**Relation to example difficulty**. Equation 3 provides an interesting bridge between memorisation and *example difficulty*. For example, TracIn (Pruthi et al., 2020) and GRAD (Paul et al., 2021) also use the evolution of model predictions across training steps to identify samples that are difficult to learn; roughly, these are samples which cause large loss updates when they are trained on. Another related measure of sample difficulty is the RHO-loss (Mindermann et al., 2022), wherein the training loss is contrast with the *irreducible* loss when a sample is only present in a holdout set. The C-score also correlates with the *prediction depth* (Baldock et al., 2021), which computes model predictions at intermediate layers, and reports the earliest layer beyond which all predictions are consistent. Equation 3 is also related to the notion of *forgetting* (Toneva et al., 2019; Zhou et al., 2022): a count of the number of transitions from *learned* to *forgotten* during training an example undergoes. Another related metric is the *learning speed*: the earliest training iteration after which the model predicts the ground truth class for that example in all subsequent iterations.

**Memorisation versus generalisation**. Given that the ultimate goal of statistical learning is *generalisation*, it is natural to ask whether this is at odds with "memorisation". A classical result establishes that a lookup table as implemented by the $k$-NN algorithm is universally consistent (Stone, 1977). Interpolating models such as boosting with decision stumps have similarly been shown to generalise (Bartlett et al., 1998), and more refined analyses have been conducted for modern interpolating neural models (Bartlett et al., 2017; Dziugaite & Roy, 2017; Brutzkus et al., 2018; Belkin et al., 2018; Neyshabur et al., 2019; Liang & Rakhlin, 2020; Montanari & Zhong, 2020; Bartlett et al., 2020; Vapnik & Izmailov, 2021). Intriguingly, some recent works have established that under certain settings, "memorisation" may be *necessary* for generalisation, either in the sense of interpolation (Cheng et al., 2022), stability-based label memorisation (Feldman, 2019), or stronger example-level memorisation (Brown et al., 2021).

**Implicit versus explicit memorisation**. Memorisation has received particular interest in the context of large language models (LLMs), such as GPT (Brown et al., 2020) and T5 (Raffel et al., 2020). Here, "memorisation" typically refers to the ability of a model to recall factual information present in the training set (e.g., names of individuals) (Petroni et al., 2019; Roberts et al., 2020; Carlini et al., 2021; 2022). This aligns with the notion from statistical learning of "memorisation" as employing a lookup table. While LLMs are capable of *implicit* memorisation, several works have shown benefits from augmenting neural models with an *explicit* memorisation component (Lample et al., 2019; Guu et al., 2020; Khandelwal et al., 2020; Borgeaud et al., 2021). Similar ideas have also proven useful outside of NLP (Panigrahy et al., 2021; Vapnik & Izmailov, 2021; Wang & Shao, 2022).

| Definition | References |
|---|---|
| Zero training error | (Zhang et al., 2017) |
| Zero training error + label is random | (Arpit et al., 2017; Yao et al., 2020; Stephenson et al., 2021) |
| Training error below Bayes error rate | (Bubeck et al., 2020; Cheng et al., 2022) |
| Prediction based on spurious correlations | (Sagawa et al., 2020; Glasgow et al., 2022) |
| Ability to reconstruct from other training samples | (Radhakrishnan et al., 2020; Carlini et al., 2021) |
| Inability to predict when removed from training sample | (Feldman, 2019; Jiang et al., 2021a) |

Table 1: Summary of existing definitions of "memorisation" of a training sample. In this paper, we focus on the final row, which proposes a stability-based metric quantifying sample predictability.

## B    HYPERPARAMETER SETTINGS

Our experiments use standard ResNet-v2 (He et al., 2016) and MobileNet-v3 (Howard et al., 2019a) architectures. Specifically, for CIFAR, we consider the CIFAR ResNet-$\{110, 56, 44, 32, 20, 14, 8\}$ family of architectures; for Tiny-ImageNet, we consider the ResNet-$\{152, 101, 50, 34, 18\}$ and the MobileNet-v3 Large architecture with scale factors $\{0.35, 0.50, 0.75, 1.00, 1.25\}$. For all ResNet models, we employ standard augmentations as per He et al. (2016a).

We train all models to minimise the softmax cross-entropy loss via minibatch SGD, with hyperparameter settings per Table 2.

| Parameter | CIFAR10* | Tiny-ImageNet |
|---|---|---|
| Weight decay | $10^{-4}$ | $5 \cdot 10^{-4}$ |
| Batch size | 1024 | 256 |
| Epochs | 450 | 90 |
| Peak learning rate | 1.0 | 0.1 |
| Learning rate warmup epochs | 15 | 5 |
| Learning rate decay factor | 0.1 | Cosine schedule |
| Learning rate decay epochs | $200, 300, 400$ | N/A |
| Nesterov momentum | 0.9 | 0.9 |
| Distillation weight | 1.0 | 1.0 |
| Distillation temperature | 3.0 | 1.0 |

Table 2: Summary of training hyperparameter settings.

## C  SUMMARY OF TRAIN AND TEST PERFORMANCE WITH MODEL DEPTH

In Table 3 we report train and test accuracies across architectures on CIFAR-100 from the one-hot training, while in Table 4 we report train and test accuracies from the distillation training across teachers of varying depths. We find that increasing depth results in models that *interpolate* the training set, while also generalising better on the test set. We also show how how distillation worsens train accuracy while improving the test accuracy.

| Architecture | Train | Test |
|---|---|---|
| ResNet-8 | 0.66 | 0.57 |
| ResNet-14 | 0.82 | 0.65 |
| ResNet-20 | 0.90 | 0.67 |
| ResNet-32 | 0.98 | 0.69 |
| ResNet-44 | 1.00 | 0.71 |
| ResNet-56 | 1.00 | 0.71 |
| ResNet-110 | 1.00 | 0.73 |

Table 3:  Train and test accuracies across architectures on CIFAR-100. Consistent with a growing body of work, on the clean dataset we find that increasing depth results in models that *interpolate* the training set, while also generalising better on the test set.

(a) ResNet-110 teacher.

| Architecture | Train | Test |
|---|---|---|
| ResNet-110→ResNet-8 | 0.65 | 0.59 |
| ResNet-110→ResNet-14 | 0.68 | 0.58 |
| ResNet-110→ResNet-20 | 0.78 | 0.64 |
| ResNet-110→ResNet-32 | 0.95 | 0.73 |
| ResNet-110→ResNet-44 | 0.98 | 0.74 |
| ResNet-110→ResNet-56 | 0.99 | 0.75 |

(b) ResNet-56 teacher.

| Architecture | Train | Test |
|---|---|---|
| ResNet-56→ResNet-8 | 0.67 | 0.60 |
| ResNet-56→ResNet-14 | 0.81 | 0.69 |
| ResNet-56→ResNet-20 | 0.88 | 0.71 |
| ResNet-56→ResNet-32 | 0.94 | 0.74 |
| ResNet-56→ResNet-44 | 0.97 | 0.75 |
| ResNet-56→ResNet-56 | 0.98 | 0.75 |

Table 4:  Train and test accuracies across architectures on CIFAR-100 under distillation averaged over 5 runs. We find that, compared to the model of corresponding architecture trained using the one-hot objective, distillation lowers train accuracy and in most cases improves test accuracy.

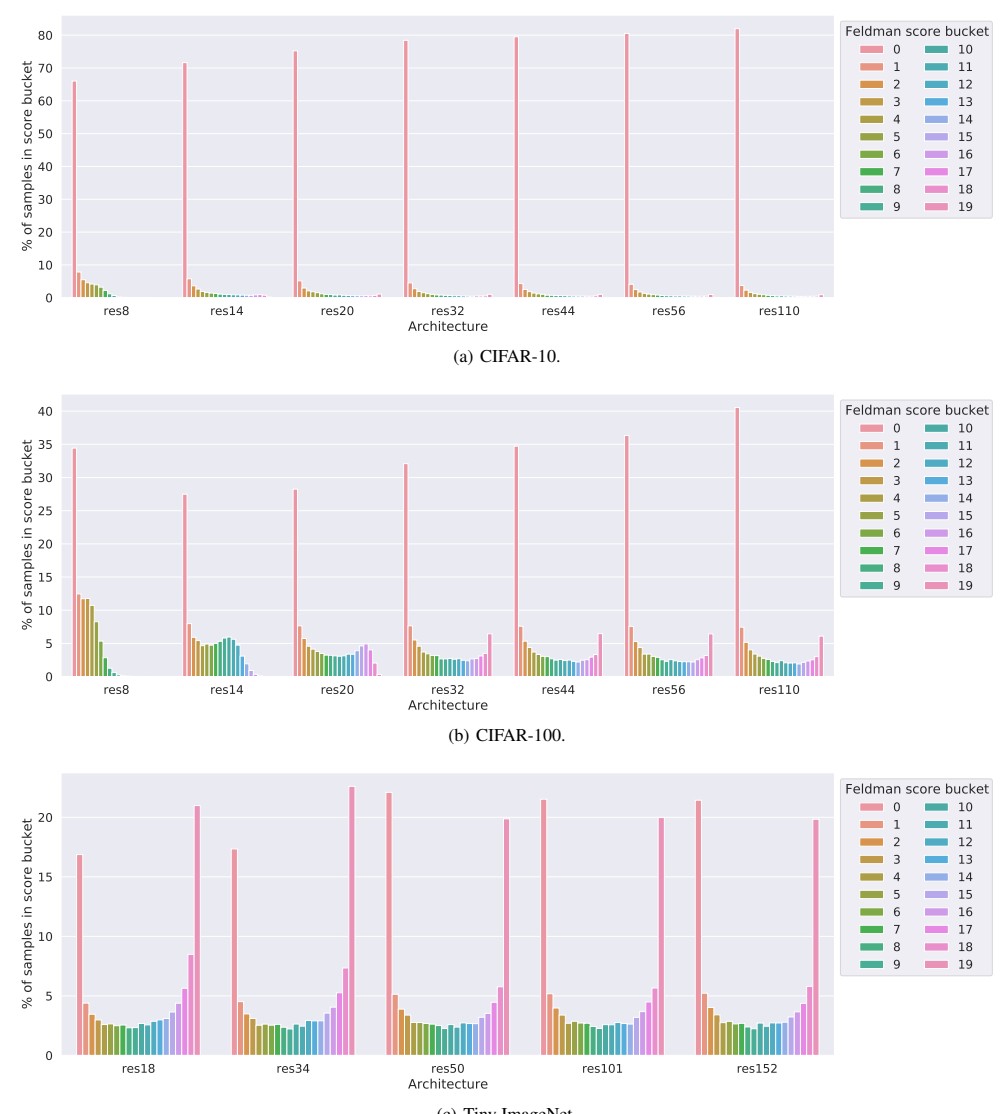

(a) CIFAR-10.

(b) CIFAR-100.

(c) Tiny ImageNet.

Figure 6: Distributions of memorisation scores across datasets and ResNet model depths. A general trend is that the memorisation scores are *bi-modal*, with most samples' score being close to 0 or 1. Further, this bi-modality is *exaggerated with model depth*: larger models (unsurprisingly) assign low score ("generalise") on relatively more samples, *but* also (more surprisingly) assign high score ("memorise") on relatively more samples as well.

## D  ADDITIONAL EXPERIMENTS: STABILITY-BASED MEMORISATION

In Figure 6 we show the histogram of memorisation scores as a function of model depth. As in the body, we see that increasing depth has the effect of exaggerating the bi-modality of the scores. Figure 7 shows a similar plot on CIFAR-100, where we vary the *width* of a ResNet-32 model.

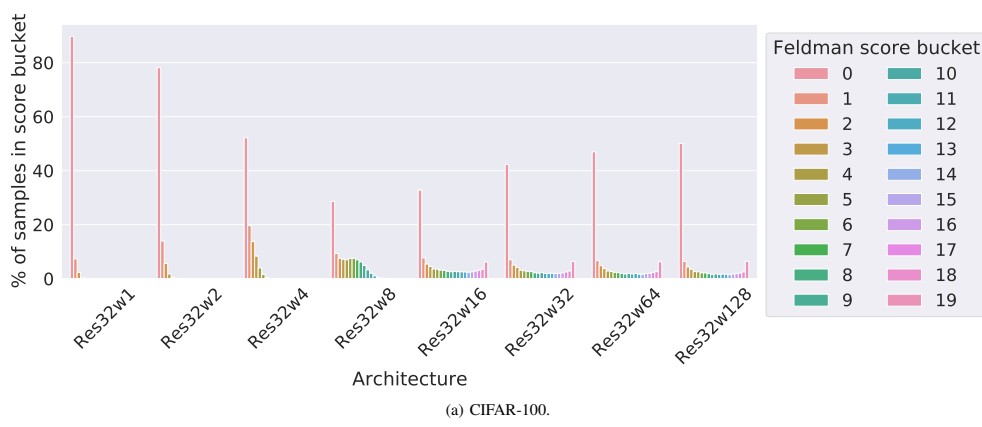

(a) CIFAR-100.

Figure 7: Distributions of memorisation scores across model *widths* of a ResNet-32 on CIFAR-100. A general trend is that the memorisation scores are *bi-modal*, with most samples' score being close to $0$ or $1$. Further, this bi-modality is *exaggerated with model width*: larger models (unsurprisingly) assign low score ("generalise") on relatively more samples, *but* also (more surprisingly) assign high score ("memorise") on relatively more samples as well.

### D.1 ADDITIONAL EXPERIMENTS: PER-EXAMPLE TRAJECTORIES OF MEMORISATION OVER MODEL CAPACITY

In Figures 8,10 we show additional examples across the three changes in memorisation criteria: least changing, most increasing and least decreasing in memorisation as the model capacity increases. In Tables 5, 9, we show predictions from ResNet-20 and ResNet-110 for each example depicted across the Figures. In the discussion below, we refer to examples from Figure 8.

We find that predictions for the least changing memorisation examples are saturated in the correct class, demonstrating that these are easy across model architectures. Predictions for the examples with lowering memorisation are assigned low probability from ResNet-20 for the correct class, and high probability from ResNet-110. We can see that these examples are often challenging and mislabeled by reasonable classes early on (e.g., the `bowl` example with a high probability for `clock` and `plate`).

Predictions for the examples with increasing memorisation are often assigned high probability from ResNet-20 to categories which are also present in the image (`willow tree` also contains `forest`, and `telephone` also contains `keyboard`). We call these *hard labeled ambigous examples*. The other type we can see are *ambigous examples*, such as `shark` and `pear`, which are not clear and easily confused with other labels as predicted by ResNet-110 (respectively, `dolphin` and `sweet pepper` labels).

We believe the presence of ambigous examples among both increasing and decreasing memorisation trajectories may be explained by the fact that the multiple labels of ambiguous points may contain labels of various difficulty, with the smaller models assigning high probability to the easy labels, while larger models shift more towards the harder labels. Then, depending on whether the easy or the hard label is present in the training set, different memorization trajectories are observed. If the observed label is the easier label, as model size gets larger, more probability gets assigned to the harder labels. Thus, increasing model size may lead to increasing memorisation trajectory. Conversely, if the observed label is the harder label, increasing model size may lead to the decreasing memorisation trajectory.

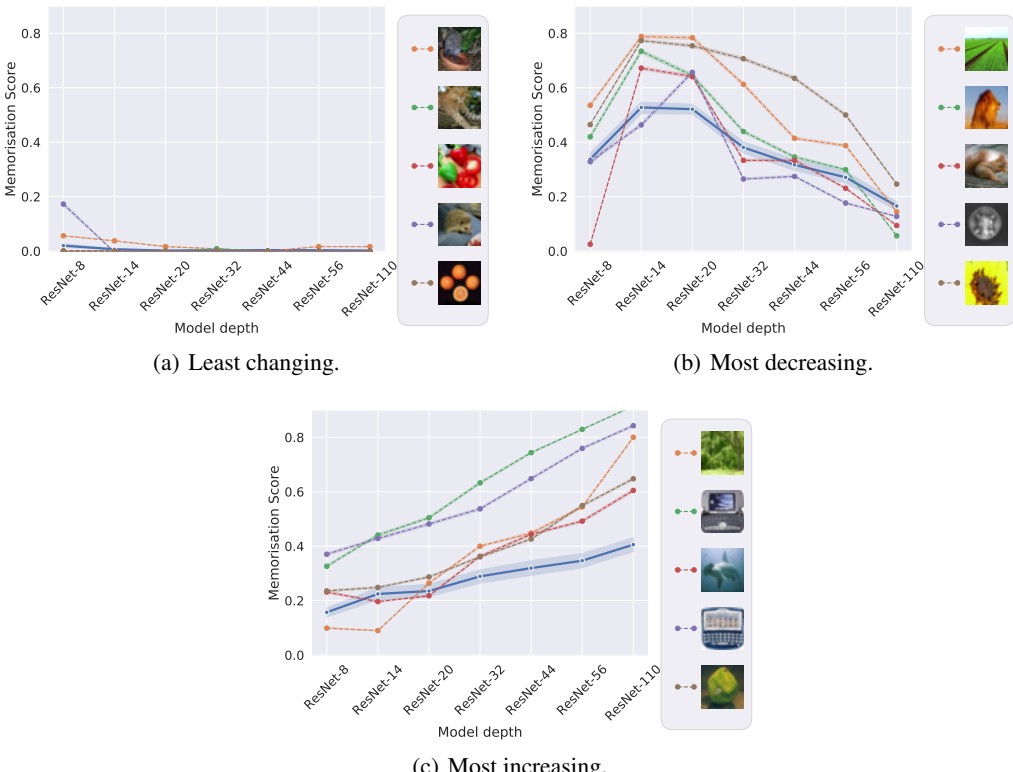

(a) Least changing.

(b) Most decreasing.

(c) Most increasing.

Figure 8: Memorisation of examples across model depths which are interpolated by a ResNet-20 model, and trajectories of memorisation for examples with: the change in memorisation closest to zero, the most decreasing memorisation, and the most increasing memorisation, when comparing ResNet-20 and ResNet-110 architectures. Solid blue line denotes average trajectory according to the top 1% example selected based on the corresponding criterion, and the dashed lines denote top 5 examples according to the corresponding criterion. We find that the fixed memorisation examples are easy and unambiguous, as is the case for `porcupine`. The decreasing memorisation example (`plain`) is arguably more complex and gets confused with `caterpillar` and `road` classes by ResNet-20. The increasing memorisation example (`willow tree`) is ambiguous (ResNet-110 predicts forest which is arguably also a valid label). In Table 5, we show predictions from ResNet-20 and ResNet-110 for each example depicted across the subfigures.

| Example | Label | ResNet-20 predictions | ResNet-110 predictions |
|---------|-------|----------------------|------------------------|
|  | porcupine | porcupine: 0.98
crab: 0.01
possum: 0.01 | porcupine: 0.98
shrew: 0.02
girl: 0.00 |
|  | leopard | leopard: 1.00
worm: 0.00
hamster: 0.00 | leopard: 1.00
worm: 0.00
hamster: 0.00 |
|  | sweet pepper | sweet pepper: 1.00
worm: 0.00
girl: 0.00 | sweet pepper: 1.00
worm: 0.00
girl: 0.00 |
|  | porcupine | porcupine: 1.00
worm: 0.00
girl: 0.00 | porcupine: 1.00
worm: 0.00
girl: 0.00 |
|  | orange | orange: 1.00
worm: 0.00
hamster: 0.00 | orange: 1.00
worm: 0.00
hamster: 0.00 |
|  | plain | caterpillar: 0.39
plain: 0.22
worm: 0.20 | plain: 0.86
caterpillar: 0.08
road: 0.02 |
|  | lion | lion: 0.36
camel: 0.16
skyscraper: 0.09 | lion: 0.94
skyscraper: 0.02
hamster: 0.01 |
|  | squirrel | squirrel: 0.36
snail: 0.27
seal: 0.21 | squirrel: 0.91
snail: 0.06
seal: 0.03 |
|  | bowl | bowl: 0.34
clock: 0.29
plate: 0.15 | bowl: 0.87
plate: 0.06
clock: 0.03 |
|  | sunflower | bee: 0.30
sunflower: 0.25
butterfly: 0.20 | sunflower: 0.75
bee: 0.10
butterfly: 0.06 |
|  | willow tree | willow tree: 0.74
forest: 0.21
palm tree: 0.03 | forest: 0.75
willow tree: 0.20
oak tree: 0.02 |
|  | telephone | telephone: 0.50
television: 0.40
keyboard: 0.06 | television: 0.88
telephone: 0.09
keyboard: 0.03 |
|  | shark | shark: 0.78
dolphin: 0.15
whale: 0.05 | shark: 0.40
dolphin: 0.32
whale: 0.21 |
|  | telephone | telephone: 0.52
keyboard: 0.37
clock: 0.08 | keyboard: 0.74
telephone: 0.16
clock: 0.09 |
|  | pear | pear: 0.71
sweet pepper: 0.21
aquarium fish: 0.03 | sweet pepper: 0.45
pear: 0.35
bowl: 0.12 |

Table 5: Predictions from ResNet-20 and ResNet-110 for each example depicted in Figure 8.

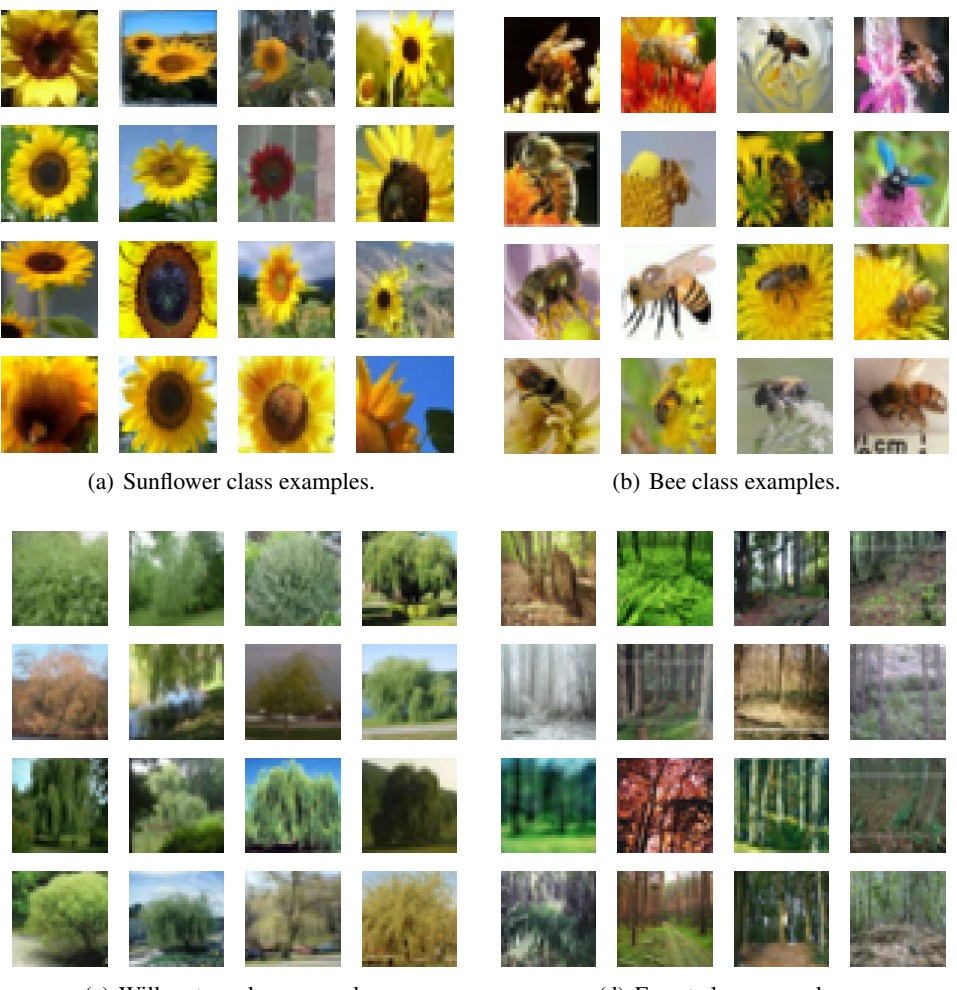

(a) Sunflower class examples.

(b) Bee class examples.

(c) Willow tree class examples.

(d) Forest class examples.

Figure 9: Randomly chosen examples from four classes from CIFAR-100 dataset: sunflower, bee, willow tree and forest.

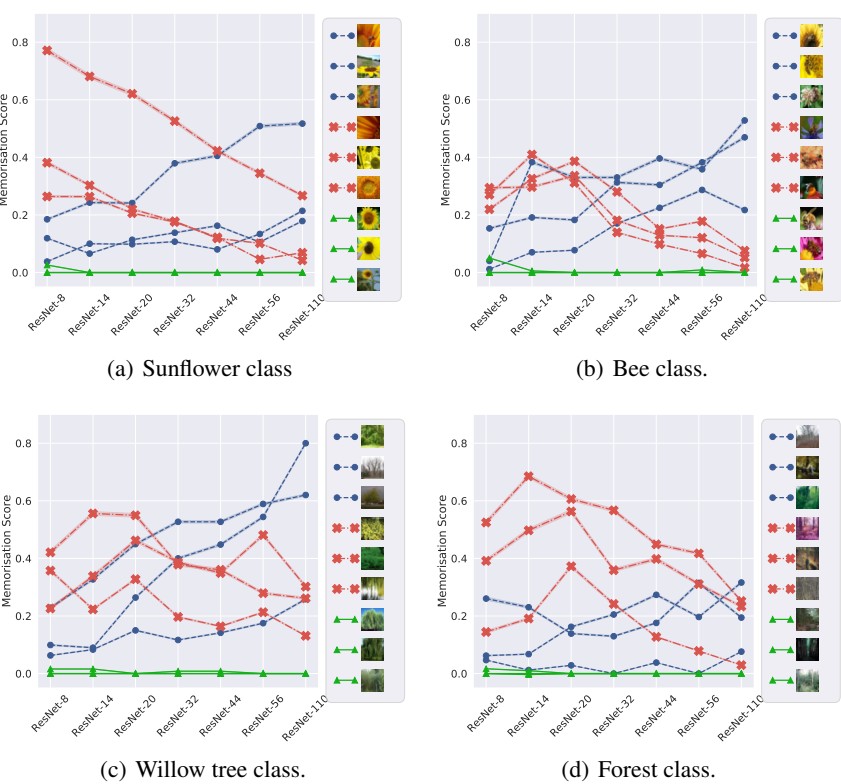

Figure 10: Memorisation of examples from a specific label across model depths which are interpolated by a ResNet-20 model, and trajectories of memorisation for examples with: the change in memorisation closest to zero, the most decreasing memorisation, and the most increasing memorisation, when comparing ResNet-20 and ResNet-110 architectures.

| Example | Label | ResNet-20 predictions | ResNet-110 predictions |
|---------|-------|----------------------|------------------------|
| | sunflower | sunflower: 0.76
bee: 0.18
poppy: 0.05 | sunflower: 0.48
bee: 0.45
poppy: 0.06 |
| | sunflower | television: 0.75
wardrobe: 0.11
can: 0.08 | television: 0.71
wardrobe: 0.12
tiger: 0.10 |
| | sunflower | willow tree: 0.29
forest: 0.27
maple tree: 0.22 | willow tree: 0.53
forest: 0.41
maple tree: 0.05 |
| | sunflower | sunflower: 0.38
sweet pepper: 0.20
rose: 0.06 | sunflower: 0.73
sweet pepper: 0.09
pear: 0.03 |
| | sunflower | sunflower: 0.71
lobster: 0.14
crab: 0.10 | sunflower: 1.00
worm: 0.00
girl: 0.00 |
| | sunflower | sunflower: 0.70
bicycle: 0.07
spider: 0.05 | sunflower: 0.96
bicycle: 0.02
man: 0.01 |
| | sunflower | sunflower: 1.00
worm: 0.00
girl: 0.00 | sunflower: 1.00
worm: 0.00
girl: 0.00 |
| | sunflower | sunflower: 1.00
worm: 0.00
girl: 0.00 | sunflower: 1.00
worm: 0.00
girl: 0.00 |
| | sunflower | sunflower: 1.00
worm: 0.00
girl: 0.00 | sunflower: 1.00
worm: 0.00
girl: 0.00 |

Table 6: Predictions from ResNet-20 and ResNet-110 for each example from Figure 10.

| Example | Label | ResNet-20 predictions | ResNet-110 predictions |
|---------|-------|----------------------|------------------------|
|  | bee | mouse: 0.31
squirrel: 0.17
possum: 0.14 | mouse: 0.44
squirrel: 0.27
rabbit: 0.08 |
|  | bee | rocket: 0.56
lizard: 0.11
cloud: 0.09 | rocket: 0.44
cloud: 0.21
lizard: 0.09 |
|  | bee | lamp: 0.25
can: 0.23
telephone: 0.19 | can: 0.36
telephone: 0.25
lamp: 0.17 |
|  | bee | bee: 0.70
shrew: 0.15
beetle: 0.05 | bee: 1.00
worm: 0.00
house: 0.00 |
|  | bee | bee: 0.71
wolf: 0.05
tiger: 0.04 | bee: 0.95
wolf: 0.01
shrew: 0.01 |
|  | bee | bee: 0.74
crab: 0.10
beetle: 0.06 | bee: 0.95
cockroach: 0.03
beetle: 0.02 |
|  | bee | bee: 1.00
worm: 0.00
house: 0.00 | bee: 1.00
worm: 0.00
house: 0.00 |
|  | bee | bee: 1.00
worm: 0.00
house: 0.00 | bee: 1.00
worm: 0.00
house: 0.00 |
|  | bee | bee: 0.98
caterpillar: 0.01
squirrel: 0.01 | bee: 0.98
caterpillar: 0.02
worm: 0.00 |

Table 7: Predictions from ResNet-20 and ResNet-110 for each example from Figure 10.

| Example | Label | ResNet-20 predictions | ResNet-110 predictions |
|---|---|---|---|
|  | willow tree | oak tree: 0.65
maple tree: 0.34
pine tree: 0.01 | oak tree: 0.64
maple tree: 0.35
pine tree: 0.01 |
|  | willow tree | oak tree: 0.67
maple tree: 0.32
pine tree: 0.01 | oak tree: 0.87
maple tree: 0.13
hamster: 0.00 |
|  | willow tree | oak tree: 0.37
maple tree: 0.36
pine tree: 0.27 | oak tree: 0.41
maple tree: 0.35
pine tree: 0.24 |
|  | willow tree | willow tree: 0.81
forest: 0.12
pear: 0.02 | willow tree: 0.95
forest: 0.04
bottle: 0.01 |
|  | willow tree | willow tree: 0.46
sunflower: 0.16
forest: 0.16 | willow tree: 0.61
forest: 0.32
caterpillar: 0.03 |
|  | willow tree | willow tree: 0.81
forest: 0.10
pine tree: 0.05 | willow tree: 0.94
forest: 0.05
cloud: 0.01 |
|  | willow tree | willow tree: 0.98
forest: 0.02
worm: 0.00 | willow tree: 0.98
forest: 0.02
worm: 0.00 |
|  | willow tree | willow tree: 1.00
worm: 0.00
hamster: 0.00 | willow tree: 1.00
worm: 0.00
hamster: 0.00 |
|  | willow tree | willow tree: 1.00
worm: 0.00
hamster: 0.00 | willow tree: 1.00
worm: 0.00
hamster: 0.00 |

Table 8: Predictions from ResNet-20 and ResNet-110 for each example from Figure 10.

| Example | Label | ResNet-20 predictions | ResNet-110 predictions |
|---------|-------|----------------------|------------------------|
|  | forest | road: 1.00
worm: 0.00
girl: 0.00 | road: 1.00
worm: 0.00
girl: 0.00 |
|  | forest | train: 0.76
streetcar: 0.18
pine tree: 0.02 | train: 0.82
streetcar: 0.16
tank: 0.02 |
|  | forest | spider: 0.32
lizard: 0.21
dinosaur: 0.13 | spider: 0.51
lobster: 0.17
table: 0.12 |
|  | forest | forest: 0.80
wardrobe: 0.11
skyscraper: 0.02 | forest: 0.98
wardrobe: 0.02
worm: 0.00 |
|  | forest | forest: 0.86
dinosaur: 0.04
crab: 0.02 | forest: 0.98
bridge: 0.01
tractor: 0.01 |
|  | forest | forest: 0.90
bridge: 0.03
house: 0.03 | forest: 0.99
table: 0.01
worm: 0.00 |
|  | forest | forest: 1.00
worm: 0.00
hamster: 0.00 | forest: 1.00
worm: 0.00
hamster: 0.00 |
|  | forest | forest: 1.00
worm: 0.00
hamster: 0.00 | forest: 1.00
worm: 0.00
hamster: 0.00 |
|  | forest | forest: 1.00
worm: 0.00
hamster: 0.00 | forest: 1.00
worm: 0.00
hamster: 0.00 |

Table 9: Predictions from ResNet-20 and ResNet-110 for each example from Figure 10.

| Example | Label | ResNet-20 predictions | ResNet-110 predictions |
|---|---|---|---|
|  | sunflower | sunflower: 0.76
bee: 0.18
poppy: 0.05 | sunflower: 0.48
bee: 0.45
poppy: 0.06 |
|  | sunflower | television: 0.75
wardrobe: 0.11
can: 0.08 | television: 0.71
wardrobe: 0.12
tiger: 0.10 |
|  | sunflower | willow tree: 0.29
forest: 0.27
maple tree: 0.22 | willow tree: 0.53
forest: 0.41
maple tree: 0.05 |
|  | sunflower | sunflower: 0.38
sweet pepper: 0.20
rose: 0.06 | sunflower: 0.73
sweet pepper: 0.09
pear: 0.03 |
|  | sunflower | sunflower: 0.71
lobster: 0.14
crab: 0.10 | sunflower: 1.00
worm: 0.00
girl: 0.00 |
|  | sunflower | sunflower: 0.78
bowl: 0.09
poppy: 0.07 | sunflower: 0.97
sweet pepper: 0.03
worm: 0.00 |
|  | sunflower | sunflower: 1.00
worm: 0.00
girl: 0.00 | sunflower: 1.00
worm: 0.00
girl: 0.00 |
|  | sunflower | sunflower: 1.00
worm: 0.00
girl: 0.00 | sunflower: 1.00
worm: 0.00
girl: 0.00 |
|  | sunflower | sunflower: 1.00
worm: 0.00
girl: 0.00 | sunflower: 1.00
worm: 0.00
girl: 0.00 |

Table 10: Predictions from ResNet-20 and ResNet-110 for each example from Figure 1.

# E ADDITIONAL EXPERIMENTS: DISTILLATION

Figure 11 shows how memorisation distribution changes under distillation across architectures and datasets. Consistently with the one-hot training depicted in Figure 6 we see both the high memorisation and low memorisation points to increase in number as architecture becomes deeper.

In Figure 12 we plot trajectories of memorisation under distillation for examples depicted in Figure 1. We find that memorisation is overall lowered for the challenging and ambigous examples.

In Figure 13 we report additional results for how the memorisation scores change under distillation. We confirm the observations from 1(b) that distillation inhibits memorisation, especially for the highly memorised examples.

Figure 14 studies the effect of varying the teacher model used for distillation. Interestingly, the choice of teacher does not have a strong influence on the results, with even the self-distillation setting for a ResNet-32 resulting in an inhibition of memorisation.

We inspect examples where distillation changes memorisation the most compared to the one-hot model. In Figure 15 we contrast memorisation trajectories across examples where memorisation is most *increased* by distillation, most *decreased*, and where it *least changes*. We find that distillation reduces memorisation for hard and ambiguous examples, whereas the remaining groups of examples are easy and unambiguous. Notice how examples where distillation increases memorisation mostly yield similar trajectories to where distillation does not impact memorisation.

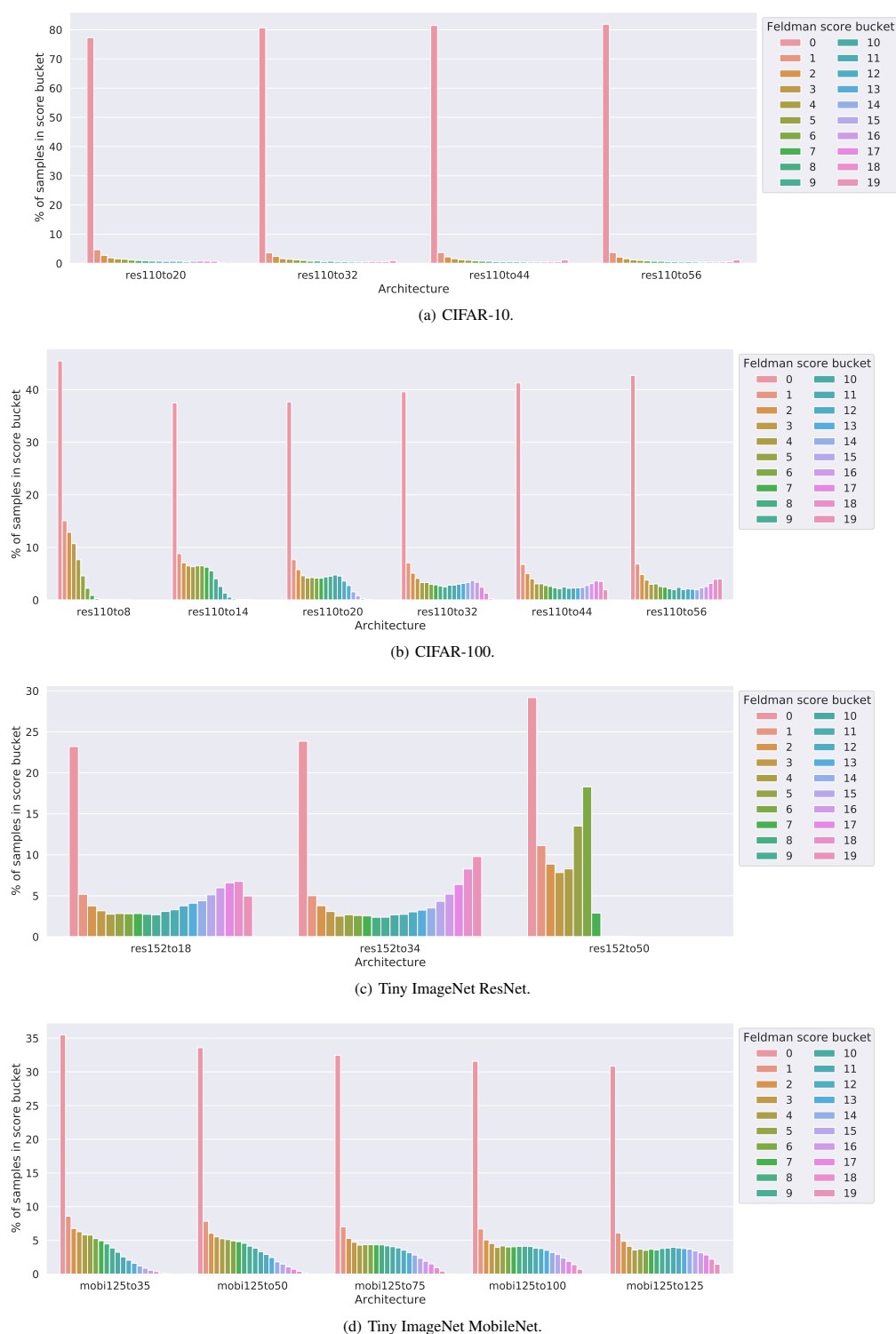

(a) CIFAR-10.

(b) CIFAR-100.

(c) Tiny ImageNet ResNet.

(d) Tiny ImageNet MobileNet.

Figure 11: Distributions of memorisation scores under knowledge distillation across datasets (different subfigures) and architectures (different bar colors in each subfigure). Similarly as under one-hot training, memorisation scores are *bi-modal*, and this bi-modality is *exaggerated with model depth*.

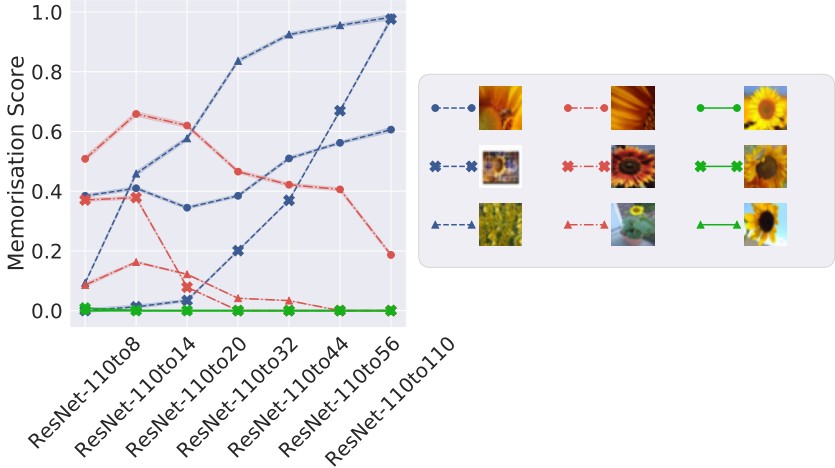

Figure 12: Illustration of how distillation affects how memorisation (in the sense of Equation 1) evolves with ResNet model depth on CIFAR-100 for examples depicted in Figure 1. We find that memorisation is overall lowered for the challenging and ambigous examples.

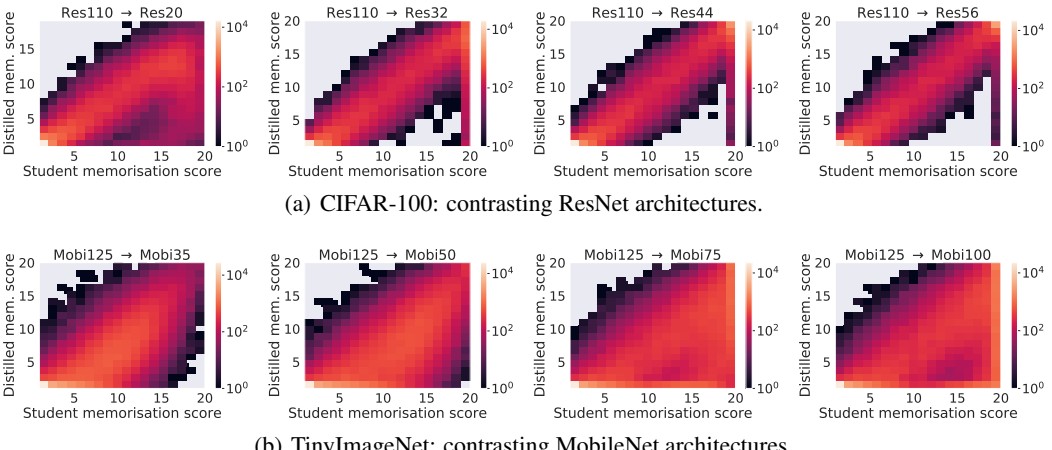

(a) CIFAR-100: contrasting ResNet architectures.

(b) TinyImageNet: contrasting MobileNet architectures.

Figure 13: Contrasting per-example memorisation scores across distilled and one-hot student models in different setups.

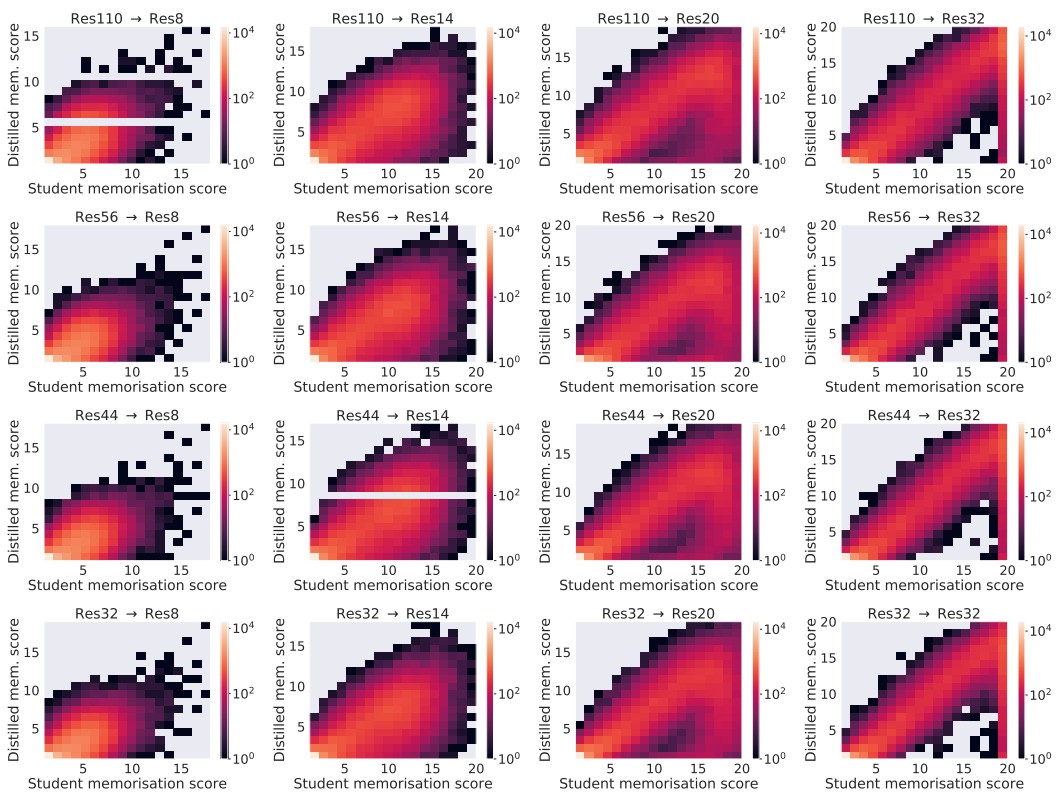

Figure 14: Contrasting per-example memorisation scores across distilled and one-hot student models for various teachers.

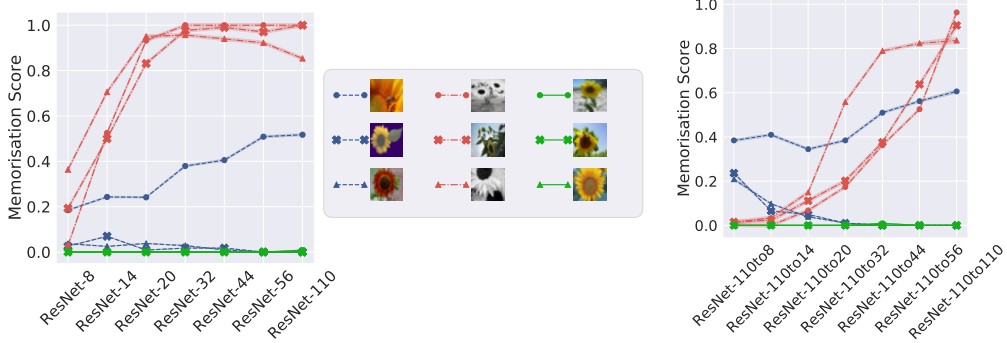

Figure 15: Evolution of memorisation score across one-hot (left) and distilled (right) models for examples across groups: where memorisation is most *increased* by distillation (blue), most *decreased* by distillation (red), and where it *least changes* (green). Distillation reduces memorisation most for hard and ambiguous examples. The remaining groups of examples are easy and unambiguous. A version of this Figure with standard deviations is shown in Figure 16 (Appendix).

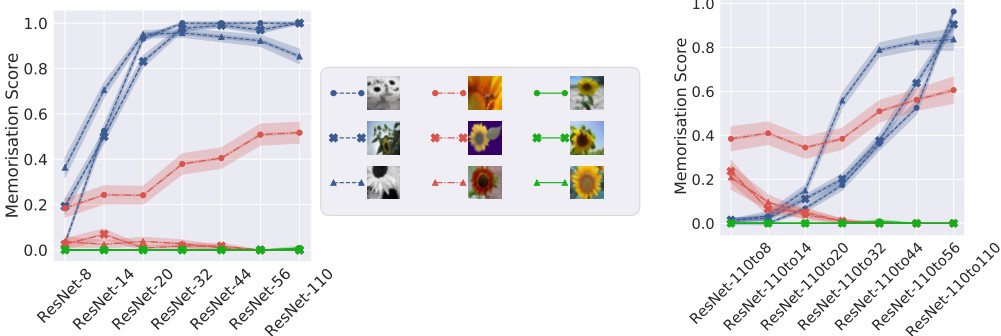

Figure 16: Evolution of memorisation score across one-hot (left) and distilled (right) models for examples across three groups: where memorisation is most *increased* by distillation (blue), most *decreased* by distillation (red), and where it *least changes* (green). We plot standard deviations of the Feldman score estimates around the example trajectories. A version of this Figure with no standard deviations for clarity is shown in Figure 15.

## F  ADDITIONAL EXPERIMENTS: PREDICTION DEPTH AND CPROX

Baldock et al. (2021) make a choice between KNN and linear probes. In Appendix E the authors notice how linear probes lead to all train examples having prediction depth equal to 0, leading to a trivial solution. Therefore, the authors settle on the KNN probes, which don't provide a trivial solution even on the train set. However, the authors consider only 3 architectures, with the strongest model being ResNet-18, and the largest dataset CIFAR-100.

Our goal is to consider larger scale experiments from Baldock et al. (2021) in terms of both the architectures and datasets, for which KNN probes on large embeddings quickly become computationally prohibitive. In order to allieviate the computational complexity, we consider linear probes on embeddings after average pooling, leading to a non trivial distribution of score values across examples. One supporting argument for this approach is that in ResNet architecture, the embedding after the final layer is subjected to average pooling before passing to the classification layer. Thus, our approach to linear probes is consistent with how ResNet does classification.

In Figure 17 we show the comparison between KNN and linear probes on average pooled embeddings. We see how there is a significant degree of similarity between the two distribution, except the low end of depths where the KNN probes assign more examples. Notice how the distribution of score values for KNN probes (leftmost figure) resembles that from Baldock et al. (2021) (see Figure 1 in Baldock et al. (2021)), despite the latter being computed on non-average pooled embeddings, contrary to the former.

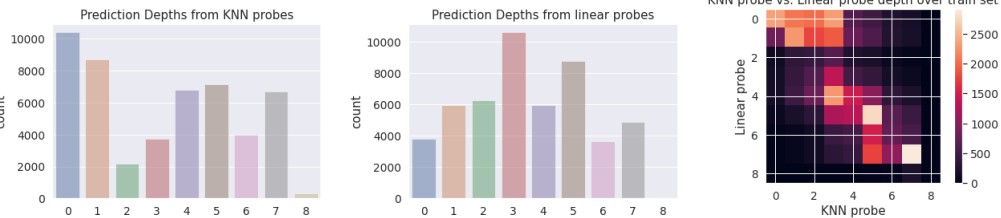

Figure 17: Prediction depth statistics on train examples from CIFAR-10 for ResNet-18 across linear or KNN probes on average pooled embeddings after each layer. Notice how linear probes lead to fewer examples with prediction depth 0 compared to KNN probes (two left-most figures). We notice how there is a high agreement between linear probes and KNN probes (right-most figure), except where KNN probes assign depth 0 or 1, where linear probes increases depth by up to 3, and in the medium range of depths, where linear probes increase the depth by up to 2.

In Figure 18 we report additional results on how prediction depth and c-score proxy scores change per example across model depths. We confirm that the shift in score values is smaller than that for memorisation score (see Figure 4).

In Figure 20 we report prediction depth score trajectories over model depths for examples shown in Figure 1. We find that all examples vary less in prediction depth scores compared to the memorisation scores.

**Different trajectories over model depths.**  In Figure 20, we plot trajectories of prediction depth over architecture depths for examples depicted in Figure 1. Recall that in the latter, we found a range of patterns depending on the relative change of memorisation score. For prediction depth, we find that the least changing in memorisation points get the lowest depth across architectures, which may be interpreted as classifying them as the easiest. Interestingly, the most and the least changing examples in terms of their memorisation score are not clearly distinguished between when considering prediction depth: most of them get assigned a very high prediction depth scores across architectures.

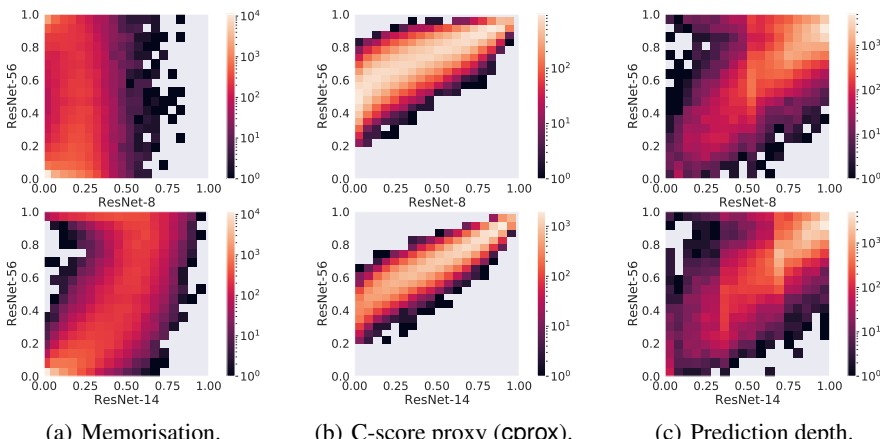

(a) Memorisation.   (b) C-score proxy (cprox).   (c) Prediction depth.

Figure 18: Contrasting per-example stability-based memorisation, prediction depth and **cprox** scores across architectures in different setups. Notice how **cprox** plot is concentrated around the diagonal to a greater extent than memorisation score. Prediction depth is also more concentrated around the diagonal, albeit to a lesser extent than **cprox** (notice there is more mass towards top left part of the heatmap for prediction depth than for **cprox**, but less than for memorisation score). Overall, the plots suggest that there are relatively few samples whose prediction depth or **cprox** score changes significantly with increased depth. By contrast, a non-negligible fraction of samples receive a low stability-based memorisation score under a ResNet-8 model, but a much higher score under a ResNet-56 model.

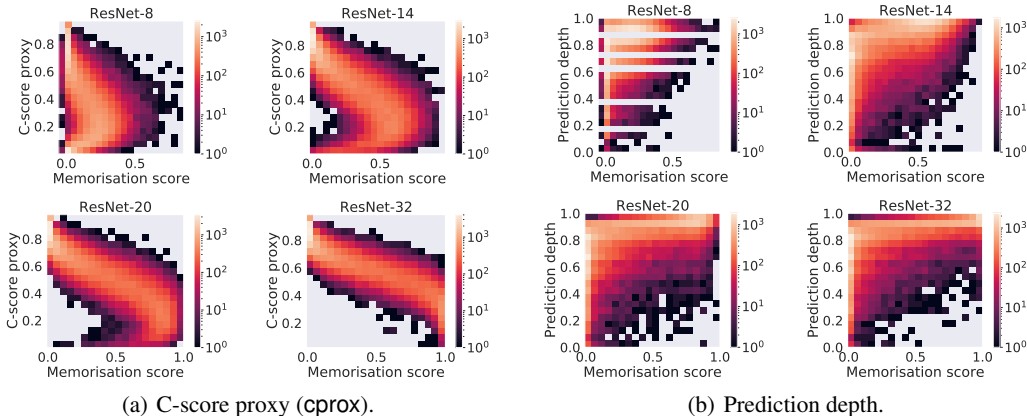

(a) C-score proxy (cprox).       (b) Prediction depth.

Figure 19: Relationship between stability-based memorisation (Equation 1) and alternate measures of memorisation on CIFAR-100. Each plot shows a heatmap of the joint density under the two measures. There is a strong correlation with stability-based memorisation, particularly for the C-score proxy: the correlation to the memorisation score across examples is above 90% for all compared model depths, while the correlation between prediction depth and memorisation score is above 70%.

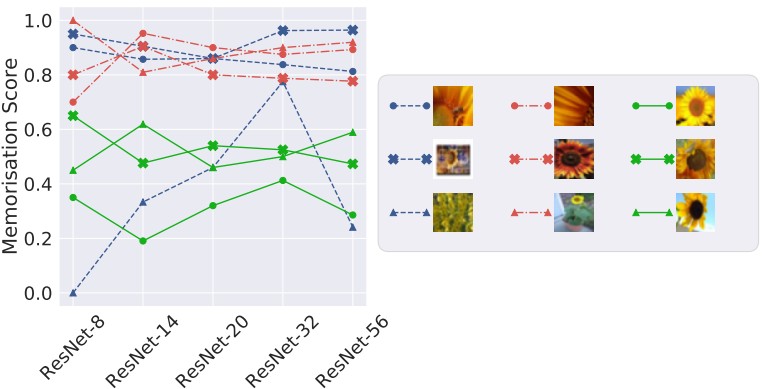

Figure 20: Prediction depth score trajectories for the examples from Figure 1. We find that all examples become less changing in their score when considering prediction depth compared to memorisation score: their trajectory of score over architecture depths is usually roughly constant. This aligns with the heatmap in Figure 18 where see that across various architecture depths the example prediction depth usually doesn't change much.

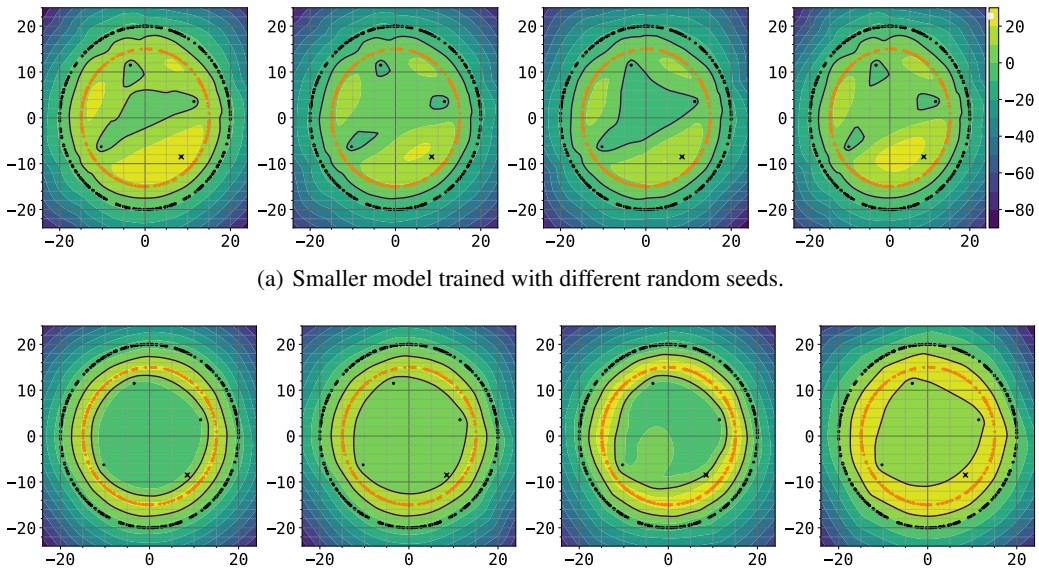

(a) Smaller model trained with different random seeds.

(b) Larger model trained with different random seeds.

Figure 21: We show the learnt decision regions (i.e., the difference between the logits from the model assigned to the two classes) when training a smaller (1 hidden layer model with 10k dimensions) and a larger (3 hidden layers with 500 dimensions) models across multiple random seeds on an illustrative two-dimensional binary classification dataset when *excluding* an outlier (denoted by the black cross). We observe how the larger model is more robust than the smaller model, as denoted by the learnt decision regions being more smooth, and the inner circle being connected as opposed to the disconnected areas learnt by the small model. The memorisation score for the highlighted example from the smaller model is $1.0$ and from the larger model is $0.65$.

## G  CONNECTION OF MEMORISATION TO ROBUSTNESS

To illustrate the connection between memorisation and robustness, we consider a two-dimensional toy binary classification dataset with two classes denoted by black points (class $0$) and orange points (class $1$). In Figure 21, we visualise the learnt decision regions when training a small (1 hidden layer model with 10000 dimensions) and a large (3 hidden layers with 500 dimensions) models across multiple random seeds when *excluding* an outlier (denoted by the black cross). Concretely, we show the difference between logits returned by the model for the two classes; large positive (negative) values denote areas where the model is confident about the class being positive (negative).

We observe that the large model learns smooth decision regions, with the inner circle being connected, as opposed to the disconnected areas learnt by the small model. Thus, intuitively, the larger model is more robust than the smaller model. At the same time, when calculating the stability-based memorisation score for the highlighted example, for the smaller model we obtain $1.0$ and for the larger model we obtain $0.65$. This hints that with greater robustness, one can expect lower memorisation of samples.

In order to measure the robustness difference between the two models more rigorously, we randomly perturb the highlighted outlier with random Gaussian corruptions of varying standard deviations and compare the accuracy on the perturbed example against the original label across the smaller and larger models (Cohen et al., 2019). We plot the decision regions of the compared models and the result of the robustness comparison in Figure 22, and find the larger model to be significantly more robust.

To summarise, on the challenging example, we find the larger model to be more robust, while having lower memorisation.

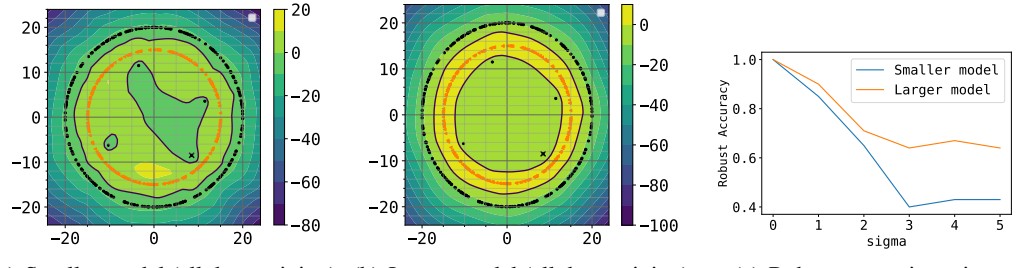

(a) Smaller model (all data training). (b) Larger model (all data training). (c) Robustness trajectories.

Figure 22: We show the learnt decision regions (i.e., the difference between the logits from the model assigned to the two classes) when training a smaller (1 hidden layer model with 10k dimensions) and a larger (3 hidden layers with 500 dimensions) models on an illustrative two-dimensional binary classification dataset. Note that contrary to Figure 21, here we *include* the highlighted outlier (denoted by the black cross). We observe how the larger model is more robust than the smaller model on the highlighted outlier, as denoted by the higher accuracy from the larger model on perturbed inputs with random Gaussian corruptions of varying standard deviations (Cohen et al., 2019).

