# OpenReview forum: "What do large networks memorize?"
_ICLR.cc/2023/Conference — Submitted to ICLR 2023_

### Official Review · Reviewer_E5os · 2022-10-23

**Confidence:** 5
**Correctness:** 4
**Technical Novelty And Significance:** 3
**Empirical Novelty And Significance:** 4
**Recommendation:** 6

**Clarity, Quality, Novelty And Reproducibility:**

## Clarity
The clarity of the paper is very good. The figures are usually clear but in some cases (pointed in the summary below) has confusing line styles and no legends.

## Novelty and Originality.

The paper builds of prior work, which is not a drawback but rather a plus. However, I think the paper can improve its novelty and originality by also discussing its impact on other topics like sparsity, privacy and robustness as discussed below.

**Strength And Weaknesses:**

## Strengths
* Understanding the impact and diversity of memorisation in modern deep neural networks is an important topic for both understanding neural networks as well as for various notions of reliability of neural networks.
* The paper is easy to read and does a good job of balancing the mathematical notations and textual verbosity.
* The authors ask specific questions and designs experiments to validate or negate their hypothesis. I find this approach of empirical validation or falsification quite well done.

## Weaknesses
While the paper provides an interesting empirical study on a fine-grained understanding of memorisation in deep neural networks,  I think the paper lacks in two ways (I illustrate them further in the summary below)
 * Several works have now linked memorisation to other properties like sparsity, robustness, and privacy. I think it would be good to see how the outcomes of this study relate to this. If the authors think a comprehensive study is outside the focus, I think, there should at least be a discussion with some preliminary results. This will not only inspire future works but also increase the relevance of this work.
* An empirical study where depth and architecture are pain axes of variation, the authors should discuss something more than ResNet and MobileNet. Even all the Mobilenet results have been relegated to the appendix. I think the main paper should include those but in addition, another relevant architecture type should be added.


**Summary Of The Paper:**

The paper discusses memorisation of training examples in deep neural networks.
* In particular, the paper shows how the memorisation profile of an architecture varies according to the depth of the networks.  The paper also shows what kind of examples are more or less memorised in deeper and shallower models.
* The paper discusses how this is impacted by distillation and carries out a similar study, as above, for distilled models.
* Finally, the paper studies how computationally efficiently computable metrics of memorisation vary from the exact but intractable one.

**Summary Of The Review:**

## Main points

* The argument that _"While the study of memorisation in large neural models is fascinating, its practical relevance is stymied by a basic fact: such models are typically inadmissible for real-world settings with constraints on latency and memory. Instead, such models are typically compressed via distillation (Bucila et al., 2006; Hinton et al., 2015)"_ is rather weird and does not help with the motivation of the paper. Further, the citations provided are also not very relevant for models that are deployed in the real world today.


* One of the motivations of the model is the work of Feldman et. al. 2019 , whose main argument is that in certain distributions memorisation is necessary for generalisation. However, this paper also mentioned _"Consequently, a natural hypothesis is that memorisation of individual training samples might initially increase (as a weak model may have insufficient capacity to memorise), but should eventually saturate or reduce (as more expressive models generalise better)."_ is a contradiction of that argument and is thus, in some way, self contradicting.

* The observation that memorisation score decreases for larger models is an interesting observation. I find Figure 2(a) and 2(b) to be an excellent illustration of this.
Does this possibly explain why adversarial robustness of deeper models is better ? For example, this is observed in Appendix B in Madry et. al. and Paleka et. al and the references therein argues that memorisation of (noisy) training data provably leads to higher adversarial error. I think this and other kinds of similar study on the impact of this observation would make a very nice addition to this paper.

* Figure 1(a) is also a good illustration of the fact that there are different kinds of examples that are memorised. Do the _increasing_ examples correspond to the human noisy labels as observed in Wei et. al ?  If that is the case, that deeper models do not need to memorise (or not as much) noisy labels, then it must also have impact on private learning as well as learning with compressed models with them ? For example, Sanyal et. al. show that private models are more inaccurate examples with higher memorisation score and similarly Hooker et. al. for compressed models but not talk about what kind of "high memorisation" scores they are inaccurate on whether _increasing_ or _decreasing_. I think this would be a great addition to this study. Similar questions also apply for adversarial training as observed in Xu et. al 2021.

* All the mathematical metrics here are expectations of random variables where the expectation is over the randomness of the training algorithm. I would like to see what is the empirical variance as well. The colour map, if I understand correctly, shows the variance across examples but not the variance due to the randomness in the algorithm.

* As this is an empirical study, I find it to be limiting to only discuss ResNet and MobileNet. It would be nice if the paper should also discuss pure convolutional architectures at least if not transformer architectures to make the study more comprehensive.

* That distillation inhibits overall memorisation and particularly those that display a higher memorisation score in non-distilled score is not very surprising but a nice thing to know. Perhaps a connection can be drawn to why distillation helps in adversarial robustness.

I find the paper interesting and would like to see the above changes made or at least argued why they should not be. Upon that, I am willing to increase my score.

## Minor points

* I am a bit confused my the line styles employed in Fig 3(a). Is there a particular reason for the way it is ?
* Figure 6 is missing the colour bar.
* More generally, I think the figures could be a bit smaller and more interesting results from the appendix could be brought to the main text. For example Figure 14

Madry et al. "Towards deep learning models resistant to adversarial attacks." (2017).  https://openreview.net/pdf?id=rJzIBfZAb

Paleka et. al. "A law of adversarial risk, interpolation, and label noise."  (2022).  https://arxiv.org/abs/2207.03933

Wei et al. "Learning with noisy labels revisited: A study using real-world human annotations." (2021).  https://arxiv.org/pdf/2110.12088v2.pdf

Sanyal et. al. " How unfair is private learning?" (2022) https://proceedings.mlr.press/v180/sanyal22a.html.

Hooker et al. "What do compressed deep neural networks forget?." (2019). https://arxiv.org/pdf/1911.05248.pdf

Xu, Han, et al. "To be robust or to be fair: Towards fairness in adversarial training." (2021). https://proceedings.mlr.press/v139/xu21b.html

Papernot et al. "Distillation as a defense to adversarial perturbations against deep neural networks." (2016) . https://ieeexplore.ieee.org/document/7546524


================== post rebuttal ==================


The author's answers were satisfactory. The paper provides very useful insights and I lean towards acceptance at this stage.

---

> ### Author Response · Authors · 2022-11-19
> **Response to reviewer E5os (1/4)**
>
> We thank the reviewer for finding the paper interesting and expressing the willingness to increase the score upon addressing the comments. Please let us know if you have any comments to our responses.
>
> > As this is an empirical study, I find it to be limiting to only discuss ResNet and MobileNet. It would be nice if the paper should also discuss pure convolutional architectures at least if not transformer architectures to make the study more comprehensive.
>
> > Even all the Mobilenet results have been relegated to the appendix. I think the main paper should include those but in addition, another relevant architecture type should be added.
>
> Thanks for this suggestion. To address the reviewer’s comment, we moved MobileNet results to the main body and ran additional experiments using convolution neural networks of varying model widths (based on the **LeNet-5** model), showing both in the new Figure 3 in the main body. Consistent with other results we find for LeNet-5 models an increasingly bi-modal distribution of memorisation score values as the model size increases, denoting a growing number of low-memorised and high-memorised examples.
>
> > All the mathematical metrics here are expectations of random variables where the expectation is over the randomness of the training algorithm. I would like to see what is the empirical variance as well. The colour map, if I understand correctly, shows the variance across examples but not the variance due to the randomness in the algorithm.
>
> Thanks for raising this point. The variance of memorisation score due to the randomness in the algorithm is indeed of importance, and the reviewer’s understanding is correct that we didn’t explicitly report it in the paper. To address the reviewer’s comment, we show standard deviations of the Feldman score estimates around example trajectories of memorisation in the new Figure 16 (Appendix). To clarify, in the memorisation score computation the randomness comes from both the random seed of the algorithm (i.e. weight initialization, ordering of the train data) and what subsets of the data are sampled for calculations of the in-sample and out-of-sample accuracy terms, as shown in Equation 2.
>
> > One of the motivations of the model is the work of Feldman et. al. 2019 , whose main argument is that in certain distributions memorisation is necessary for generalisation. However, this paper also mentioned "Consequently, a natural hypothesis is that memorisation of individual training samples might initially increase (as a weak model may have insufficient capacity to memorise), but should eventually saturate or reduce (as more expressive models generalise better)." is a contradiction of that argument and is thus, in some way, self contradicting.
>
> We would like to clarify that the theory from (Feldman, 2019) states that *interpolation* of train examples (i.e., fitting to the train data) may be necessary for generalisation. On the other hand, our hypothesis pertains to *memorisation* decreasing for models of increasing capacity which fully interpolate the train set. Our hypothesis spans from the expectation that for models which fully interpolate the train set, as generalisation increases memorisation may decrease. Notice that on average, we do find this simple hypothesis to be true.
>
> > The argument that "While the study of memorisation in large neural models is fascinating, its practical relevance is stymied by a basic fact: such models are typically inadmissible for real-world settings with constraints on latency and memory. Instead, such models are typically compressed via distillation (Bucila et al., 2006; Hinton et al., 2015)" is rather weird and does not help with the motivation of the paper. Further, the citations provided are also not very relevant for models that are deployed in the real world today.
>
> Thanks for this suggestion. The citations we provided are, to our knowledge, the canonical references for knowledge distillation. We do agree that it is useful to have more recent citations that explicitly detail the use of distillation in modern models deployed in practice. We have added citations to that end to Section 4 in the revised version.

---

> > ### Author Response · Authors · 2022-11-19
> > **Response to reviewer E5os (2/4)**
> >
> > > Several works have now linked memorisation to other properties like sparsity, robustness, and privacy. I think it would be good to see how the outcomes of this study relate to this. If the authors think a comprehensive study is outside the focus, I think, there should at least be a discussion with some preliminary results. This will not only inspire future works but also increase the relevance of this work.
> >
> > Thanks for this suggestion. Indeed some works looked at memorization and its connections to sparsity, robustness, and privacy at a more aggregate level. In this work, we put memorization under a magnifying lens. Certainly, it makes sense to revisit the connections of memorization with the aforementioned property at a similar level. However, we believe this would be a significant undertaking in itself, and would be best served by a separate, future study.
> >
> > We added a discussion of the works the reviewer recommended in Section 6 of the revised version, and replied to specific comments from the review below.
> >
> > To address this comment, we also add a visual toy experiment (Appendix G) that sheds novel insight into the connection between memorisation and robustness. We add more details about it below.

---

> > > ### Author Response · Authors · 2022-11-19
> > > **Response to reviewer E5os (3/4)**
> > >
> > > > Figure 1(a) is also a good illustration of the fact that there are different kinds of examples that are memorised. Do the increasing examples correspond to the human noisy labels as observed in Wei et. al ? If that is the case, that deeper models do not need to memorise (or not as much) noisy labels, then it must also have impact on private learning as well as learning with compressed models with them ? For example, Sanyal et. al. show that private models are more inaccurate examples with higher memorisation score and similarly Hooker et. al. for compressed models but not talk about what kind of "high memorisation" scores they are inaccurate on whether increasing or decreasing. I think this would be a great addition to this study. Similar questions also apply for adversarial training as observed in Xu et. al 2021.
> > >
> > > Thanks for pointing out these very interesting works referring to related concepts to memorisation.
> > >
> > > Please note that many of the previous works use the term “memorisation” but use a different definition for it. In particular, among the references: Wei et. al define memorisation as the confidence in the label, and Hooker et. al. consider performance on the test set where the compressed model disagrees with the original model. The stability based memorisation definition instead enjoys the formal connection to generalisation as shown in [1], and it has been acclaimed by multiple works as a canonical memorisation score, e.g. [2, 3, 4].
> > >
> > > From preliminary qualitative analyses, we found that examples with increasing memorization to often be the noisy and multi-labeled examples (e.g. see Section D.1 in the Appendix), which aligns with observations made by Wei et al. regarding what kind of examples are mislabeled by annotators. It will be interesting to extend the work from Wei et al. by analysing noisy examples through the lens of the stability based memorisation.
> > >
> > > One could draw some connection to Sanyal et al., since they look at how varying privacy affects the model behaviour, and privacy is known to be connected to memorization, as pointed out by Feldman [1]. However, **there's a key difference between our works: to vary privacy levels, Sanyal et al., needed to vary their guarantee across all samples** simultaneously. In contrast, our study instead observes memorisation behavior across examples that arises as we vary the model size.
> > >
> > > In more detail, Sanyal et al. explored the discrepancy among model accuracies on different subpopulations (i.e., **fairness**) due to the privacy requirements. For the CIFAR-10 dataset the authors indeed utilised the stability-based memorisation score (referred to as **self-influence scores**) to identify the minority vs. majority subpopulations. More interestingly, as their main contribution, the authors look at fairness while varying the privacy guarantee (in a differentially private algorithm). It’s worth pointing out that Feldman [1] has noted a strong connection between the model's memorisation behavior and its differential privacy guarantee. Essentially, this implies that varying privacy guarantees is akin to forcefully changing the model’s memorisation behavior. However, note that the privacy guarantee needs to simultaneously hold for **each example**.
> > >
> > > In contrast, our work doesn’t explicitly enforce such uniform guarantees for all examples, and instead observes memorisation behavior across examples that arises as we vary the model size. That being said, studying a more nuanced (e.g., on a per-example level) connection between privacy and memorization is certainly an interesting future direction.
> > >
> > > [1] Feldman. “Does Learning Require Memorization? A Short Tale about a Long Tail”
> > >
> > > [2] Carlini et al. “Extracting Training Data from Large Language Models”
> > >
> > > [3] Baldock et al. “Deep learning through the lens of example difficulty”
> > >
> > > [4] Jiang et al. “Characterizing Structural Regularities of Labeled Data in Overparameterized Models“

---

> > > > ### Author Response · Authors · 2022-11-19
> > > > **Response to reviewer E5os (4/4)**
> > > >
> > > > > That distillation inhibits overall memorisation and particularly those that display a higher memorisation score in non-distilled score is not very surprising but a nice thing to know. Perhaps a connection can be drawn to why distillation helps in adversarial robustness.
> > > >
> > > > > The observation that memorisation score decreases for larger models is an interesting observation. I find Figure 2(a) and 2(b) to be an excellent illustration of this. Does this possibly explain why adversarial robustness of deeper models is better ? For example, this is observed in Appendix B in Madry et. al. and Paleka et. al and the references therein argues that memorisation of (noisy) training data provably leads to higher adversarial error. I think this and other kinds of similar study on the impact of this observation would make a very nice addition to this paper.
> > > >
> > > > Thanks for highlighting this connection. Indeed, we show how distillation reduces memorisation, and the previous work from Papernot et al. shows how distillation improves adversarial robustness. This, together with the work from Madry et al., indeed point at the possibility of a potential connection existing between adversarial robustness and memorisation. We added a discussion into Section 6 of the revised paper about this point.
> > > >
> > > > Moreover, in Section G in the Appendix in the revised paper we add a toy example to illustrate our intuition of how adversarial robustness of a model may vary with memorisation scores from an algorithm. In particular, we find that adversarially robust models tend to produce decision boundaries which can better generalise to even challenging examples excluded from the train set. Thus, algorithms generating less adversarially robust models yield higher memorisation scores.
> > > >
> > > > > I am a bit confused my the line styles employed in Fig 3(a). Is there a particular reason for the way it is ?
> > > >
> > > > Thanks for catching this, we now fixed the line styles. In the new plot, line styles and markers both distinguish between in-sample accuracy, out-sample accuracy and memorisation score values, whereas colours distinguish between distillation and one-hot models.
> > > >
> > > > > Figure 6 is missing the colour bar.
> > > >
> > > > Thanks for catching this, we have now added the colour bars.
> > > >
> > > > > More generally, I think the figures could be a bit smaller and more interesting results from the appendix could be brought to the main text.
> > > >
> > > > Thanks for the suggestion. To address it, we moved Figure 7 from the Appendix, which now became the new Figure 3 to additionally strengthen the results in the main paper.

---

> > > > > ### Comment · Reviewer_E5os · 2022-11-23
> > > > > **Thanks again**
> > > > >
> > > > > Thanks again for addressing my comments and for implementing the changes.

---

> > > > ### Comment · Reviewer_E5os · 2022-11-23
> > > > **Please include in the paper**
> > > >
> > > > Thank you for this discussion. I personally found this very useful and interesting and think it would be a shame to leave it on openreview and not add to the paper. There has indeed been different definitions of memorisation in different work and given that this paper takes a deep dive into this phenomenon, it would be usefil if it also discussed the different uses of this term.
> > > >
> > > > May I suggest the authors to include this total discussion in the paper as well ?

---

> > > > > ### Author Response · Authors · 2022-11-23
> > > > > **We will include the discussion from our response in the final revision**
> > > > >
> > > > > Thanks for finding our response useful and interesting. Per reviewer's suggestion, we will include this discussion in the final revision of the paper (unfortunately the openreview tool does not allow for revisions at this time).

---

> > ### Comment · Reviewer_E5os · 2022-11-23
> > **Thanks for the added experimental settings**
> >
> > Thank you for your response and the new experimental results and the clarifications.

---

### Official Review · Reviewer_wKAi · 2022-10-24

**Confidence:** 4
**Correctness:** 4
**Technical Novelty And Significance:** 1
**Empirical Novelty And Significance:** 2
**Recommendation:** 5

**Clarity, Quality, Novelty And Reproducibility:**

**Quality**

The work is very well-written and easy to follow. Plots are of a high quality and simple to understand. The Appendix contains even more numerical evaluations, making this a very complete empirical study.

**Clarity**
I am listing a few points in the following which need further clarification:
1. The authors compare the stability-based metric against other, faster memorization estimators and conclude that those metrics are not precise enough to capture the trends found by the former. The authors implicitly assume that the stability-based metric is the superior estimator (other metrics should exhibit the same trends) without giving a complete reasoning as to why this is the case. A better explanation would be helpful for the reader to understand why it is not the case that the other metrics "might be right” instead.
2. In Figure 2.a), we can see the in-sample accuracy plotted as a function of model depth. I am a bit surprised that the ResNets models of depth 8, 14 and 20 cannot reach in-sample accuracy 1, which basically is the training accuracy right? Is this standard, especially since the training set is of smaller size here, if I understood correctly?

**Originality**

As already mentioned in the "Weaknesses" section, this work largely builds on top of [1,2] but scales the empirical experiments to a larger scale. Moreover, similar findings are obtained but now re-confirmed in a broader setting. Some aspects are new such as the distillation setting and the fact that some samples are memorized less with increasing capacity.

**Strength And Weaknesses:**

**Strengths**

1. This work provides a very broad empirical study regarding memorization in the clean label setting, which has not been performed to this degree in prior work. The results obtained in this paper might help in guiding theory to better understand the phenomenon.

2. It is very interesting that other, simpler memorization measures cannot capture the trends that the stability-inspired metric exhibits. The observed trends might serve as sanity checks and thus faciliate the design of new, computationally lighter memorization measures.

**Weaknesses**

1. While this work makes very interesting observations, a large part of the findings and the methodology have already been presented in prior works [1, 2]. The metric used in this work has been developped by [1] and initial empirical results have already been obtained on large
﻿networks such as ResNets in [2]. [2] already establishes that samples with the highest memorization scores are typically ambiguous or mislabeled, while this work builds on top of this and shows that this effect is exacerbated with depth. While some novel findings are made (i.e. that some samples are memorized less under higher capacity), large parts seem to re-confirm what has been observed in the literature.

2. A novel aspect in this work is the study of knowledge distillation, which was not considered in previous works. While certainly useful in practical settings, I am not convinced whether the understanding of memorization in this specific setting is of particular relevance. Given the fact that distilled models usually generalize better than standard ones, it does not seem so surprising that less memorization happens. However, it would be interesting to understand how memorization is reduced, but the authors  do not address this question.

[1] Does learning require memorization? A short tale about a long tail. CoRR, abs/1906.05271, 2019.
[2] Vitaly Feldman and Chiyuan Zhang. What neural networks memorize and why: Discovering the long tail via influence estimation

**Summary Of The Paper:**

This work studies how and to what degree deep networks perform memorization of samples in the clean label setting. The authors build upon previous work and employ the stability-based memorization score developped in [1] which measures how much the prediction of a
model on a given sample changes if the sample is present versus not present in the training dataset. Using this measure, the authors
perform a in-depth study on how memorization is affected as a function of depth of the model as well as the effects of distillation. The
authors find that larger depth generally decreases memorization on average, which is in-line with the fact that deeper models generalize
better. Interestingly, a more local look reveals a more fine-grained picture: There are samples that are less memorized as depth increases
but at the same time, there are examples that are memorized more. These examples often correspond to mislabeled samples. Moreover,
the authors find that distillation of a larger teacher into a smaller student reduces memorization on average.

**Summary Of The Review:**

While some novel results are obtained in this work, it still feels more like a larger empirical study that mostly re-confirms what prior work has identified. I am not convinced of the relevance of the distillation experiments since no real insight into the workings of the method are obtained (i.e. why does a distilled network memorize less). The result that simpler memorization metrics cannot capture the same trends is interesting and certainly relevant to the community. Still, in total the novel contributions of this work are not enough in my opinion and I hence recommend the rejection of this work.

---

> ### Author Response · Authors · 2022-11-19
> **Response to reviewer wKAi (1/2)**
>
>
> Thanks for the feedback. We are glad that the reviewer found our paper well-written and easy to follow, and found many empirical findings interesting. Below, we aim to respond to the reviewer’s concerns about the novelty and significance of our contributions. As per the reviewer’s suggestion, we have also included additional results and a discussion on the effect of distillation on memorisation. We hope that our responses will prompt the reviewer to reconsider their assessment of our submission.
>
>
> > While this work makes very interesting observations, a large part of the findings and the methodology have already been presented in prior works [1, 2]. The metric used in this work has been developped by [1] and initial empirical results have already been obtained on large ﻿networks such as ResNets in [2]. [2] already establishes that samples with the highest memorization scores are typically ambiguous or mislabeled, while this work builds on top of this and shows that this effect is exacerbated with depth. While some novel findings are made (i.e. that some samples are memorized less under higher capacity), large parts seem to re-confirm what has been observed in the literature.
>
> We wish to emphasise that prior work evaluated memorisation on *individual* ResNet models, as we note in the Introduction. However, a systematic analysis of how memorisation is affected by model size and distillation, and whether efficient proxies of memorization capture all aspects of memorization, has been lacking. Such an analysis is precisely the main contribution of our work.
>
> Note that many of our conclusions are not a-priori obvious. For example, when seeking an answer to the question how memorisation changes with a growing model capacity, on a population level we find the increasingly bi-modal distribution of memorisation, while on a per-sample level we find examples with both increasing and decreasing memorisation. We did not expect these findings a-priori.
>
> Thus, we respectfully disagree that “...large parts seem to **re-confirm** what has been observed in the literature.”
>
>
> > A novel aspect in this work is the study of knowledge distillation, which was not considered in previous works. While certainly useful in practical settings, I am not convinced whether the understanding of memorization in this specific setting is of particular relevance. Given the fact that distilled models usually generalize better than standard ones, it does not seem so surprising that less memorization happens.
>
> We agree that knowledge distillation is known to improve generalisation (e.g., as we note in Table 3). However, from this it is not clear a-priori how distillation should impact memorisation. For example, one could wonder whether the improved generalisation would come due to the lowered memorisation of the otherwise highly memorised examples, or whether the memorisation reduction would be uniform across all examples.
>
> > However, it would be interesting to understand how memorization is reduced, but the authors do not address this question.
>
> We agree that finding the reasons for memorisation decreasing under distillation would be of great interest for the community. Please note that explaining why distillation improves generalisation is a related effort, with a number of papers trying to address it (e.g., even amongst the submissions to this year’s ICLR), but no work definitively answers this question yet. Similarly, understanding why memorisation decreases in distillation is a worthwhile and ambitious effort. In our work, we make first steps in this direction by showing that distillation can indeed lower memorisation in the first place.
>
> In an attempt to address the reviewer’s question, we provide further insights into how distillation leads to lower memorisation in the revised version: we expand distillation analysis by adding and discussing the new Figure 15 in Appendix E outlining examples and their memorisation trajectories where distillation most affects memorisation. To summarise the new result, we find that distillation reduces memorisation for hard and ambiguous examples, whereas the remaining groups of examples are easy and unambiguous.

---

> > ### Author Response · Authors · 2022-11-19
> > **Response to reviewer wKAi (2/2)**
> >
> > > The authors compare the stability-based metric against other, faster memorization estimators and conclude that those metrics are not precise enough to capture the trends found by the former. The authors implicitly assume that the stability-based metric is the superior estimator (other metrics should exhibit the same trends) without giving a complete reasoning as to why this is the case. A better explanation would be helpful for the reader to understand why it is not the case that the other metrics "might be right” instead.
> >
> > We agree that depending on the end goal, one could argue for different definitions of memorisation. We provide a review of different notions of memorisation and difficulty in Appendix A. In our work we focused on the stability-based notion of memorisation as it enjoys a formal connection to generalisation as shown in [1], and it has been acclaimed by multiple works as a canonical memorisation score, e.g. [2, 3, 4]. We then considered the specific proxies based on the papers that introduced them claiming how they highly relate to the stability based memorisation score [3, 4]. Thus, even the works that introduce these proxies consider the stability-based memorisation score as the gold-standard.
> >
> > [1] Feldman. “Does Learning Require Memorization? A Short Tale about a Long Tail”
> >
> > [2] Carlini et al. “Extracting Training Data from Large Language Models”
> >
> > [3] Baldock et al. “Deep learning through the lens of example difficulty”
> >
> > [4] Jiang et al. “Characterizing Structural Regularities of Labeled Data in Overparameterized Models“
> >
> > > In Figure 2.a), we can see the in-sample accuracy plotted as a function of model depth. I am a bit surprised that the ResNets models of depth 8, 14 and 20 cannot reach in-sample accuracy 1, which basically is the training accuracy right? Is this standard, especially since the training set is of smaller size here, if I understood correctly?
> >
> > The reviewer has correctly interpreted the figure. Note that these results are on CIFAR-100, which is a little more challenging on CIFAR-10. On CIFAR-10, we indeed observe that the training accuracy of even smaller ResNet models can reach 100%.

---

### Official Review · Reviewer_bHhw · 2022-10-24

**Confidence:** 3
**Correctness:** 3
**Technical Novelty And Significance:** 3
**Empirical Novelty And Significance:** 3
**Recommendation:** 6

**Clarity, Quality, Novelty And Reproducibility:**

The clarity is mostly good except for some minor issues. The quality, empirical novelty and reproducibility are good.

**Strength And Weaknesses:**

# Strength
- The writing and presentation are nice. It is a pleasure to read this paper.
- The empirical studies are very thorough. I appreciate the authors' efforts in doing so.
- The paper is very relevant to ICLR community. The paper makes a step toward understanding the memorization and generalization of deep learning.
- The findings in this paper may be useful to spark new applications in scenarios such as data selection/reweighting when learning under noise and active learning.

# Weaknesses

Generally, in the aspects of displaying findings and drawing conclusions, the paper does not have significant issues. However, I would like to see improvements regarding the following points to make the paper a better contribution to the community.
- In the caption of figure 2, you state "however for architectures starting at depth 32, only a small number of points remain with
high memorisation score values". Is the used "small" correct? I see a relatively large number of points have high scores when the depth is no less than 32.
- "More interestingly, this bi-modality is exaggerated with model depth". However, as I notice, figure 2(b) cannot reflect this finding clearly. The exaggeration is very marginal. Can this be shown more confidently in some other scenarios? E.g., on mini-imagenet.
- On page 7, by inspecting figure 1(b), you state "Interestingly, none of the examples with small memorisation score from the one-hot model obtain a significant increase in memorisation from distillation." At first, it is hard for me to clearly understand figure 1(b). Are the plotted quantities the number of data? Besides, is the argument "none of" correct?
- Why does distillation have such impacts on memorization scores? Is the found phenomenon generalizable to other settings, architectures, or modalities? More explanations are desired.
- Do the different results between the stability-based score and C-score come from that C-score omits the first term in Eq 2? But as said, the first term can be ignored when the model capacity is enough. So does the conclusion in section 5 still hold when you used larger architectures like vision transformers? I want to see a clarification on this point.
- Can you provide some extra experiments on adversarial examples? I want to know what kind of data will neural networks view the adversarial examples as？Ambigous or hard?





**Summary Of The Paper:**

The paper presents a systematic empirical analysis of the memorization effect of DNNs w.r.t. model size and distillation. It bases the studies on the existing memorization measure 'memorisation score', and performs extensive experiments. The findings are 4-fold: (1) increasing the model complexity tends to make memorization more bi-modal where some hard examples get lower memorization scores while some ambigous examples get higher memorization scores; (2) distillation aids to inhibit memorisation; (3) some other memorization measures can strongly correlate with the used stability-based memorisation measure, but they do not capture key trends seen in the latter.

**Summary Of The Review:**

This is a good paper with extensive empirical findings yet flaws in aspects of clarity and explanation. Currently, I am learning to accept this paper.

---

> ### Author Response · Authors · 2022-11-19
> **Response to reviewer bHhw (1/2)**
>
> Thanks for the feedback. We are glad you found it “a pleasure to read this paper”.
>
> > Can you provide some extra experiments on adversarial examples? I want to know what kind of data will neural networks view the adversarial examples as？Ambigous or hard?
>
> Thanks for the suggestion. We would like to clarify that to compute the stability-based memorisation score on a sample, that sample must be included as part of the training set (to assess the influence of including versus removing it during training). By contrast, to our knowledge, adversarial examples are defined given an already trained model, and are typically newly constructed examples that are not part of the training set.  It is therefore not clear how memorisation scores for model-dependent adversarial examples could be calculated.
>
> Nevertheless, it is a very good question how adversarial robustness relates to memorisation. In Section G in the Appendix in the revised paper we add a toy example to illustrate our intuition of how adversarial robustness of a model may vary with memorisation scores from an algorithm. In particular, we find that adversarially robust models tend to produce decision boundaries which can better generalise to even challenging examples excluded from the train set. Thus, algorithms generating less adversarially robust models yield higher memorisation scores.
>
> > In the caption of figure 2, you state "however for architectures starting at depth 32, only a small number of points remain with high memorisation score values". Is the used "small" correct? I see a relatively large number of points have high scores when the depth is no less than 32.
>
> Thanks for catching this. It is indeed true that even for depths higher than 32, there remain examples with high memorisation scores. To clarify, the observation we meant to emphasise in the caption is that compared with smaller depth models, while the count of highly memorised examples increases, there is an even larger increase in low memorisation examples, leading to average memorisation getting lower.  We corrected the caption in the revised version.
>
> > "More interestingly, this bi-modality is exaggerated with model depth". However, as I notice, figure 2(b) cannot reflect this finding clearly. The exaggeration is very marginal. Can this be shown more confidently in some other scenarios? E.g., on mini-imagenet.
>
> Thanks for this suggestion. Please note that in Figure 6 (Appendix) in the revised version of the paper we demonstrate the increasing bi-modality using histogram plots across multiple datasets, including on Tiny-ImageNet. We also add additional results using convolution neural networks of varying model widths (based on the **LeNet-5** model) in the new Figure 3.
>
> > On page 7, by inspecting figure 1(b), you state "Interestingly, none of the examples with small memorisation score from the one-hot model obtain a significant increase in memorisation from distillation." At first, it is hard for me to clearly understand figure 1(b). Are the plotted quantities the number of data? Besides, is the argument "none of" correct?
>
> The reviewer is correct that in Figure 1b), each cell shows the count of examples which give a certain memorisation score under one-hot training (x axis) and under distillation (y axis).
>
> We indeed can see that the area in the upper left corner of each plotted matrix shows count 0, and thus we stated that none of the examples with small memorisation score from the one-hot model obtain a significant increase in memorisation from distillation. We clarify this in the revised version of the draft in Section 4.
>
> > Why does distillation have such impacts on memorization scores? Is the found phenomenon generalizable to other settings, architectures, or modalities? More explanations are desired.
>
> We provide more results in Appendix E, in particular on Tiny-ImageNet where we contrasted the MobileNet and ResNet architectures. We find consistent results with our observations.
>
> We agree that running experiments on further settings and modalities would be of great interest. Please note the computational challenge of running experiments to compute the stability-based memorisation score: even the approximation employed as specified in Equation 2 requires training hundreds of models for any given setting of dataset and architecture.

---

> > ### Author Response · Authors · 2022-11-19
> > **Response to reviewer bHhw (2/2)**
> >
> > > Do the different results between the stability-based score and C-score come from that C-score omits the first term in Eq 2? But as said, the first term can be ignored when the model capacity is enough. So does the conclusion in section 5 still hold when you used larger architectures like vision transformers? I want to see a clarification on this point.
> >
> > Good question. We can indeed interpret the discrepancy between C-score proxy and Feldman memorisation as spanning from: 1) discrepancy between **C-score proxy** and **C-score**, 2) discrepancy between **C-score** and **Feldman memorisation score** or 3) both. The reviewer asks about the role of 2) in the discrepancy we identified.
> >
> > To answer this question, we can use Table 3 (Appendix), where we report the train accuracy numbers. There we can see that ResNet-44 and deeper ResNet models reach 100% accuracy. Therefore, for ResNet-44 and deeper the first term from Equation 2 can be indeed omitted, and Feldman memorisation becomes equal to C-score up to a constant and a sign – perfectly correlated (when M from the definition of C-score is drawn from a point-mass; see page 3 in the draft for the definition and the discussion of C-score and memorisation score).
> >
> > When comparing Figure 5a and 2b in the updated draft, we can see how for ResNet-56 the distribution of **C-score proxy** values is very different from the distribution of Feldman memorisation score values. There, **C-score** and **Feldman memorisation score** are perfectly correlated, and so the discrepancy is due to the discrepancy between **C-score proxy** and **C-score**.

---

### Author Response · Authors · 2022-11-19
**Summary of the changes**

We’d like to thank the reviewers for their considered feedback. We have uploaded a revision with the following changes (coloured in blue) incorporating this feedback:
*  A visual toy experiment (Appendix G) that sheds novel insight into the connection between memorisation and robustness (as brought up by reviewers bHhw and E5os), and the significance of examples with decreasing memorisation score (as brought up by reviewer wKAi).
* Added results with LeNet architecture in Figure 3 (as brought up by reviewers bHhw and E5os).
* Re-arranged some figures:
  * Removed Figure 2 from the original version, and replaced it with Figure 3 from the original version, since the latter mostly subsumed the former.
  * Moved Figure 7 (Contrasting per-example memorisation scores acros architectures) from the original version into the body, which is now Figure 4.
* Expanded Section 4 and Appendix F in the revised version by discussing the new Figure 15 (examples where memorisation was most affected by distillation).
* Applied fixes and improvements to the existing Figures per reviewers’ requests.
* Moved part of the proxies results from Section 5 to the appendix.

---

> ### Author Response · Authors · 2022-12-12
> **Dear reviewers**
>
> Thank you again for your detailed feedback. We have added new experiments and extensively updated the text to address your suggestions. We have also added a visual toy experiment to convey our intuitions.
>
> We sincerely look forward to your re-evaluation of the scores you've assigned to the paper. Please do not hesitate to ask any further questions.

---

### Comment · Area_Chair_N1mJ · 2022-11-20
**Question**

Hi Authors,

I'm wondering what are the possible guidance or insights that these empirical studies provide? That is, what people might do differently in the future given the empirical observations here?

Thanks,

---

> ### Author Response · Authors · 2022-11-21
> **Response to the Area Chair**
>
> Thanks for the question. The following practical implications can be drawn from our work:
>
> * **Our study points to a potentially sound way for identifying noisy examples in the labelled data.** Specifically, we find the existence of examples with increasing memorisation over model capacity even after interpolation. We qualitatively find that they are often ambiguous and mislabeled. For example, one general conclusion from this is that if the average memorisation of a subset of train data grows with model capacity, this can point at a possibility of poor quality of the labels in that set.
>   * We note that previous works offered different metrics for identifying difficult examples, and it will be an important future direction to systematically contrast memorisation against alternatives. As one step in this direction, in our work we find how the categorisation according to prediction depth (Baldock et al.) is different than according to memorisation. Indeed, contrasting Figures 1a) and Figure 20 in the updated draft, we see how the most and the least changing examples in terms of their memorisation score are not clearly distinguished when considering prediction depth: most of them get assigned a very high prediction depth scores across architectures.
>
> * **Our study points at a potential application of weighting examples based on memorisation score during distillation.** Improving distillation by reweighting or filtering examples is an active research direction in the community [1,2,3,4]. Given the finding how distillation lowers the memorisation of train examples, and how interestingly this is often achieved by lowering the interpolation (see Figure 2), this suggests that memorisation could be used for improving memorisation by filtering out or downweighting examples which the model ends up memorising.
> * **Our study leads to an important practical conclusion that one should be careful with using certain statistics as proxies for memorisation.** Previous works suggested that certain statistics based on model training or model inference (which we discuss in Section 5) can provide efficient proxies for memorisation score. Although these proxies appear to yield high correlation to memorisation, we find that distributionally they are significantly different, and do not capture the key aspects of memorization score that we uncover. This points at a future direction of identifying better memorisation score proxies which could be efficiently computed.
>
> Moreover, we would like to emphasise the importance of our findings for the broader goal of developing a better understanding of deep learning:
> * Both the increasing bi-modality of memorisation scores with increasing model capacity, and the existence of a variety of trajectories of memorisation scores with increasing model capacity, are to our knowledge new observations in the community.
> * The question of why distillation improves generalisation is still under study by the community. Our work contributes to this branch of work by studying memorisation under distillation.
>
>
>
> [1] Zhou et al. "Rethinking Soft Labels for Knowledge Distillation: A Bias-Variance Tradeoff Perspective"
>
> [2] Tang et al. "Understanding and improving knowledge distillation"
>
> [3] Lu et al. "RW-KD: Sample-wise Loss Terms Re-Weighting for Knowledge Distillation"
>
> [4] Iliopoulos et al. "Weighted Distillation with Unlabeled Examples"

---

### Decision · Program_Chairs · 2023-01-20

**Decision:**

Reject

**Justification For Why Not Higher Score:**

Weakness.

--- “While some novel findings are made (i.e., that some samples are memorized less under higher capacity), large parts seem to re-confirm what has been observed in the literature.”
--- "A novel aspect in this work is the study of knowledge distillation, which was not considered in previous works. While certainly useful in practical settings, I am not convinced whether the understanding of memorization in this specific setting is of particular relevance. "
--- Though The paper provides some form of understanding using empirical methods with some metric of memorization, there are two limitations: (1) there was no theoretical explanation for these empirical observations, (2) it's unclear what are the practical implications of these findings. The AC asks the authors the second question directly, but, unfortunately, didn't find the answers satisfactory. The authors largely seem to respond with many _potential_ applications, but there was no solid evidence that any of these implications can be eventually fleshed out.


In general, the AC finds that the paper mostly conducted systematic experiments that do not require much novelty. Given a metric for measuring memorization, arguably, one could ask many questions of the type "does factor X affect memorization" and then throw many GPUs on them. One can do this even for multiple metrics of memorization. However, it might be more valuable to study the correct interpretation and impact of memorization metrics, e.g., what is the correct memorization metric? Is memorization good or bad and should we avoid it in what situations? Can the amount of memorization in the system tell us some ways to improve the model？



**Justification For Why Not Lower Score:**

n/a

**Metareview: Summary, Strengths And Weaknesses:**

Strength: (quoted sentences are reviewers', and selected by the AC. Others are AC's own opinion.)
--- “The writing and presentation are nice. It is a pleasure to read this paper.”
--- "The empirical studies are very thorough. I appreciate the authors' efforts in doing so."
--- "It is very interesting that other, simpler memorization measures cannot capture the trends that the stability-inspired metric exhibits. The observed trends might serve as sanity checks and thus facilitate the design of new, computationally lighter memorization measures."

Weakness.

--- “While some novel findings are made (i.e., that some samples are memorized less under higher capacity), large parts seem to re-confirm what has been observed in the literature.”
--- "A novel aspect in this work is the study of knowledge distillation, which was not considered in previous works. While certainly useful in practical settings, I am not convinced whether the understanding of memorization in this specific setting is of particular relevance. "
--- Though The paper provides some form of understanding using empirical methods with some metric of memorization, there are two limitations: (1) there was no theoretical explanation for these empirical observations, (2) it's unclear what are the practical implications of these findings. The AC asks the authors the second question directly, but, unfortunately, didn't find the answers satisfactory. The authors largely seem to respond with many _potential_ applications, but there was no solid evidence that any of these implications can be eventually fleshed out.


In general, the AC finds that the paper mostly conducted systematic experiments that do not require much novelty. Given a metric for measuring memorization, arguably, one could ask many questions of the type "does factor X affect memorization" and then throw many GPUs on them. One can do this even for multiple metrics of memorization. However, it might be more valuable to study the correct interpretation and impact of memorization metrics, e.g., what are the correct memorization metrics? Is memorization good or bad and should we avoid it in what situations? Can the amount of memorization in the system tell us some ways to improve the model？